# Sustained neural rhythms reveal endogenous oscillations supporting speech perception

Sander van Bree[1,2,3], Ediz Sohoglu[1,4], Matthew H. Davis[1‡], Benedikt Zoefel[1,5,6‡*]

**1** MRC Cognition and Brain Sciences Unit, University of Cambridge, Cambridge, United Kingdom, **2** Centre for Cognitive Neuroimaging, University of Glasgow, Glasgow, United Kingdom, **3** School of Psychology and Centre for Human Brain Health, University of Birmingham, Birmingham, United Kingdom, **4** School of Psychology, University of Sussex, Brighton, United Kingdom, **5** Centre de Recherche Cerveau et Cognition, CNRS, Toulouse, France, **6** Université Toulouse III Paul Sabatier, Toulouse, France

‡ These authors are joint senior authors on this work.
* benedikt.zoefel@cnrs.fr

**Data Availability Statement:** Data and custom-built MATLAB scripts are available in the following repository: https://osf.io/xw8c4/. An additional Excel spreadsheet (S1 Data) provides the numerical data underlying summary statistics.

## Abstract

Rhythmic sensory or electrical stimulation will produce rhythmic brain responses. These rhythmic responses are often interpreted as endogenous neural oscillations aligned (or "entrained") to the stimulus rhythm. However, stimulus-aligned brain responses can also be explained as a sequence of evoked responses, which only appear regular due to the rhythmicity of the stimulus, without necessarily involving underlying neural oscillations. To distinguish evoked responses from true oscillatory activity, we tested whether rhythmic stimulation produces oscillatory responses which continue after the end of the stimulus. Such sustained effects provide evidence for true involvement of neural oscillations. In Experiment 1, we found that rhythmic intelligible, but not unintelligible speech produces oscillatory responses in magnetoencephalography (MEG) which outlast the stimulus at parietal sensors. In Experiment 2, we found that transcranial alternating current stimulation (tACS) leads to rhythmic fluctuations in speech perception outcomes after the end of electrical stimulation. We further report that the phase relation between electroencephalography (EEG) responses and rhythmic intelligible speech can predict the tACS phase that leads to most accurate speech perception. Together, we provide fundamental results for several lines of research—including neural entrainment and tACS—and reveal endogenous neural oscillations as a key underlying principle for speech perception.

## Introduction

The alignment of oscillatory neural activity to a rhythmic stimulus, often termed "neural entrainment," is an integral part of many current theories of speech processing [1–4]. Indeed, brain responses seem to align more reliably to intelligible than to unintelligible speech [5,6]. Similarly, rhythmic electrical stimulation applied to the scalp (transcranial alternating current stimulation (tACS)) is assumed to "entrain" brain oscillations and has been shown to modulate speech processing and perception [7–11]. Despite the prominence of entrainment theories in speech research and elsewhere [1,12–14], it has been surprisingly difficult to demonstrate that

**Funding:** This work was supported by the European Union's Horizon 2020 research and innovation programme under the Marie Sklodowska-Curie grant agreement number 743482 (to BZ), the British Academy/Leverhulme Trust (grant number SRG18R1\180733) (to BZ), and the Medical Research Council UK (grant number SUAG/044 G101400) (to MHD). The funders had no role in study design, data collection and analysis, decision to publish, or preparation of the manuscript.

**Competing interests:** The authors have declared that no competing interests exist.

**Abbreviations:** ECoG, electrocorticography; EEG, electroencephalography; FFT, fast Fourier transformation; HPI, head-position indicator; ICA, independent component analysis; ITC, intertrial phase coherence; LCMV, linear constrained minimum variance; MEG, magnetoencephalography; PSOLA, pitch-synchronous overlap and add; RSR, rate-specific response; SNR, signal–noise ratio; tACS, transcranial alternating current stimulation.

stimulus-aligned brain responses indeed involve endogenous neural oscillations. This is because, if each stimulus in a rhythmic sequence produces a brain response, the evoked brain responses will appear rhythmic as well, without necessarily involving endogenous neural oscillations. This is not only true for sensory stimulation: Rhythmic behavioural effects of tACS cannot be interpreted as evidence of entrained endogenous oscillations; they might simply reflect the impact of regular changes in current imposed onto the brain [15].

In the present work, we provide evidence that rhythmic intelligible speech and tACS entrain endogenous neural oscillations. Neural oscillations are often proposed to align their high-excitability phase to important events in a rhythmic sequence so as to boost the processing of these events and enhance corresponding task performance [12,13]. It is possible that such a process entails a passive, "bottom-up" component during which oscillations are rhythmically "pushed" by the stimulus, similar to the regular swing of a pendulum (that is, the endogenous oscillation is "triggered" by an exogenous stimulus). On the other hand (and not mutually exclusive), an active, "top-down" component could adjust neural activity so that it is optimally aligned with a predicted stimulus. Importantly, in both cases, we would anticipate that oscillatory brain responses are sustained for some time after the offset of stimulation: This could be because predictions about upcoming rhythmic input are upheld, and/or neural oscillations are self-sustaining and (much like a pendulum swing) will continue after the cessation of a driving input. Consequently, sustained oscillatory responses produced by a rhythmic stimulus after the cessation of that stimulus can provide evidence for entrainment of endogenous neural oscillations [16,17].

In this paper, we will contrast this theory of entrained oscillations with an alternative view in which entrainment is merely due to responses evoked directly by the stimulus per se. Note that both views are sufficient to accommodate existing evidence of brain signals aligned to a stimulus while the latter is present. Given the difficulty of distinguishing true oscillations from other responses during rhythmic input, we use the term "entrained" only to describe a signal aligned to a stimulus (irrespective of whether this alignment reflects oscillations or evoked responses; see "entrainment in the broad sense" in [14]). We then measure sustained rhythmic activity to infer its neural origins: Truly oscillatory activity that was entrained to the rhythmic stimulus would lead to sustained rhythmic responses, but sustained responses would not be expected for stimulus-evoked neural activity. In the current study, we provide 2 distinct sources of evidence for sustained oscillatory effects: (1) oscillatory MEG responses that continue after rhythmic intelligible speech and (2) oscillatory effects of tACS on speech perception that continue after the termination of electrical stimulation. Furthermore, we link these 2 effects in single participants to show how the phase of oscillatory neural responses measured with electroencephalography (EEG) can predict the tACS phase at which word report is enhanced. In combination, these findings provide evidence that endogenous neural oscillations in entrained brain responses play a causal role in supporting speech perception.

## Results

### Experiment 1: Rhythmic intelligible speech produces sustained MEG oscillations

In Experiment 1, 21 participants listened to sequences of noise-vocoded [18] rhythmic speech (Fig 1A), which were 2 or 3 seconds in duration and presented at 1 of 2 different rates (2 Hz and 3 Hz). Speech sequences consisted of 4, 6, or 9 one-syllable words, depending on sequence duration and speech rate. These words were either clearly intelligible or completely unintelligible and noise-like, depending on the number of spectral channels used during vocoding (16 or 1; see Materials and methods).

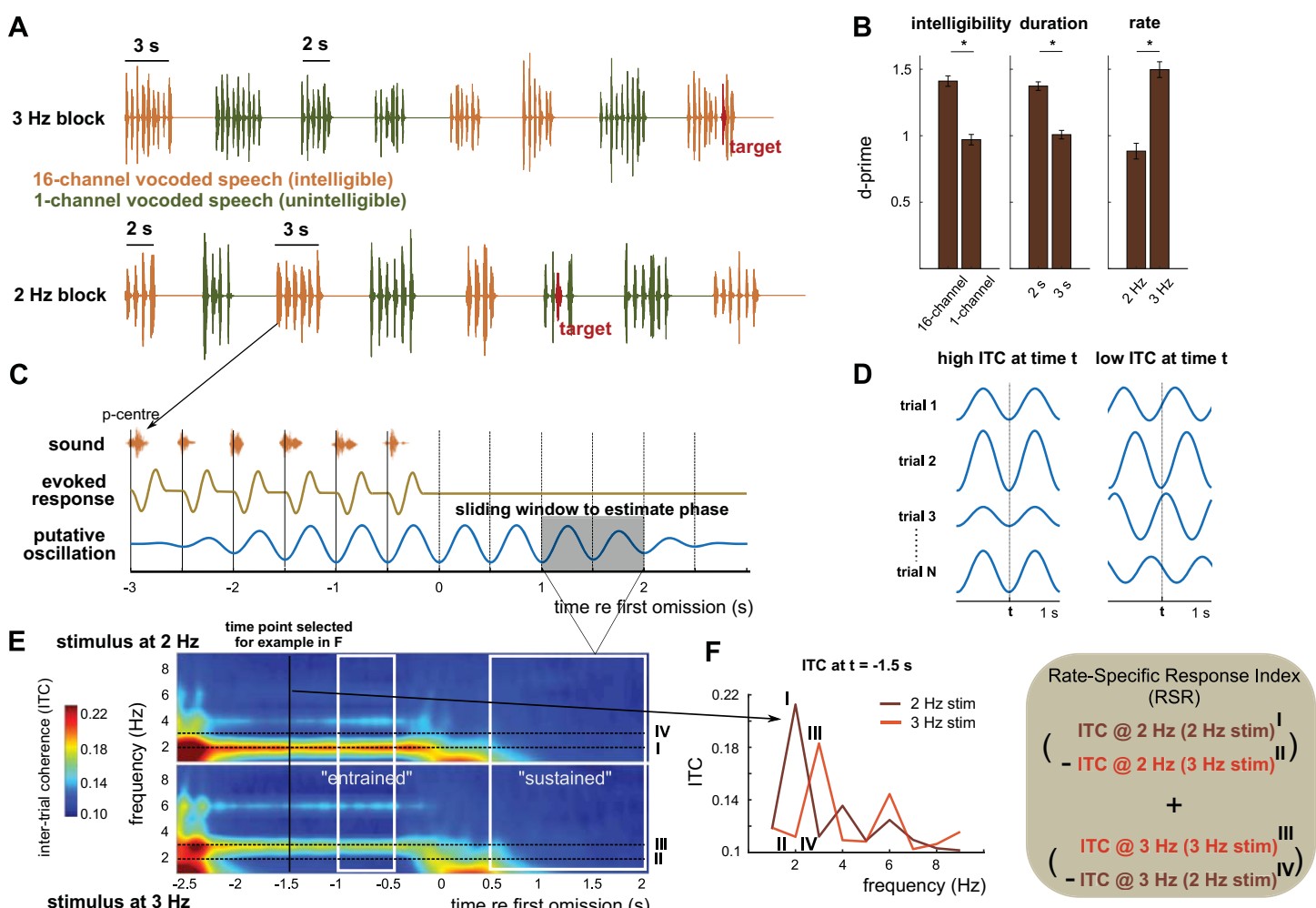

**Fig 1. Experimental paradigm and analysis.** (A) Participants listened to rhythmic speech sequences and were asked to press a button when they detected an irregularity in the stimulus rhythm (red targets). (B) Performance (as d-prime) in the irregularity detection task, averaged across participants and shown for the main effects of intelligibility, duration, and rate. Error bars show SEM, corrected for within-subject comparison [19]. Please refer to S1 Data for the numerical values underlying this figure panel. (C) A rhythmic brain response measured during the presented sounds cannot distinguish true neural oscillations aligned to the stimulus from regular stimulus-evoked responses. However, only the oscillation-based model predicts a rhythmic response which outlasts the rhythmic stimulus. For each time point t throughout the trial, oscillatory phase was estimated based on a 1-second window centred on t (shaded grey). (D) ITC at time t is high when estimated phases are consistent across trials (left) and low otherwise (right). Note that the 2 examples shown differ in their 2-Hz ITC, but have similar induced power at the same frequency. (E) ITC in the longer (3-second) condition, averaged across intelligibility conditions, gradiometers, and participants. Note that "time" (x-axis) refers to the centre of the 1-second windows used to estimate phase. ITC at 2 and 3 Hz, measured in response to 2 and 3 Hz sequences, were combined to form an RSR. The 2 time windows used for this analysis ("entrained" and "sustained") are shown in white (results are shown in Fig 2). (F) ITC as a function of neural frequency, separately for the 2 stimulation rates, and for the example time point shown as a black line in E. ITC, intertrial phase coherence; RSR, rate-specific response; SEM, standard error of mean.

In a subset of trials (12.5%), one of the words in the sequence (red in Fig 1A) was shifted towards another (± 68 ms), and participants were given the task to detect this irregularity in the stimulus rhythm. Replicating previous work [7], performance in this task (quantified as d-prime; see Materials and methods; Fig 1B) was enhanced for intelligible as compared to unintelligible speech (main effect of intelligibility in 3-way repeated-measures ANOVA, $F(1, 20) = 31.30$, $p < 0.0001$). We also found that irregularities were easier to detect if the sequence was shorter (main effect of duration, $F(1, 20) = 32.39$, $p < 0.0001$) and presented at a faster rate (main effect of rate, $F(20) = 26.76$, $p < 0.0001$; no significant interactions).

Using magnetoencephalography (MEG) and EEG, we measured brain responses during the presented sounds and, importantly, in a subsequent, silent interval of several seconds that

continued until the start of the next sequence (Fig 1A and 1C). Due to its higher signal–noise ratio, we focused our initial analyses on the MEG data. We used intertrial phase coherence (ITC) to quantify oscillatory brain responses (Fig 1D). ITC makes use of the fact that, for each of the 2 speech rates, the timing of the presented speech sequences (relative to the "perceptual centre" of individual words, vertical lines in Fig 1C) was identical across trials (see Materials and methods). ITC therefore has the advantage of directly testing the predicted temporal evolution of the recorded signal (i.e., its phase), whereas power-based measures are focused on its amplitude [20]. Fig 1E shows ITC, separately for the 2 stimulus rates, and averaged across MEG sensors and participants. For 1 example time point, Fig 1F shows ITC as a function of neural frequency.

Our hypothesis states that ITC at a given neural frequency is higher when that frequency corresponds to the stimulation rate than when it does not. For example, we expect that ITC at 2 Hz during (and after) the presentation of 2-Hz sequences (I in Fig 1E and 1F) is higher than ITC at 2 Hz during (and after) 3-Hz sequences (II in Fig 1E and 1F). By comparing ITCs across the 2 stimulus rates (I versus II and III versus IV in Fig 1E and 1F), we thus developed a precise measurement of whether brain responses follow the rate of the stimulus, which we term the rate-specific response index (RSR; see Materials and methods and formula in Fig 1F). An RSR larger than 0 indicates a brain response that is specific to the stimulus rate. Spectral measures such as ITC can be biased by other neural activity than endogenous oscillations: For example, a response caused by the omission of an expected stimulus might produce an increase in ITC that is most pronounced at low frequencies (approximately 250 ms in Fig 1E). By contrasting ITC between 2 rate conditions, RSR removes such contamination if it is independent of stimulus rate (i.e., present in both rate conditions). This property makes it—in the present case— also superior to other commonly used approaches, such as permutation tests [21,22], which would not only abolish the hypothesized rhythmic responses, but also nonrhythmic responses which produce high ITC for other reasons (e.g., evoked response to stimulus omission).

We next defined 2 time windows of interest (white in Fig 1E). The first time window ("entrained") covered the period in which sound sequences were presented while but avoiding sequence onset and offset. This period allows us to measure entrained responses (i.e., neural responses synchronised with an ongoing stimulus). A large RSR in this time window reflects a brain response aligned to the stimulus rhythm (irrespective of whether a true oscillation is involved). The other time window ("sustained") covered the silent interval between sequences while avoiding sequence offset. A large RSR in this time window is evidence for a sustained oscillatory response and, consequently, for the involvement of endogenous neural oscillations in generating stimulus-aligned entrained responses.

In the entrained time window, when averaged across all conditions, the RSR was clearly larger than 0 (cluster-based correction, $p < 0.001$; summed t = 883.39; 102 sensors in cluster), showing a typical auditory scalp topography (Fig 2A). We then contrasted the RSR across conditions (Fig 2B). We found a main effect of intelligibility (cluster-based correction, $p < 0.001$; summed t = 87.30; total of 29 sensors in 2 clusters), revealing stronger RSRs to intelligible speech in a cluster of left frontal sensors. We also found a main effect of duration, revealing a preference for shorter sequences for left frontal sensors (cluster-based correction, $p = 0.02$; summed t = −11.11; 4 sensors in cluster) and one for longer sequences for parietal sensors (cluster-based correction, $p = 0.05$; summed t = 6.83; 3 sensors in cluster). There was no significant interaction between intelligibility and duration.

Although the RSR was larger for intelligible speech, it was significantly larger than 0 (indicating the presence of an entrained response) for both intelligible (cluster-based correction, $p < 0.001$; summed t = 783.56; 102 sensors in cluster) and unintelligible speech (cluster-based correction, $p < 0.001$; summed t = 706.67; 102 sensors in cluster). Despite being reliable at all

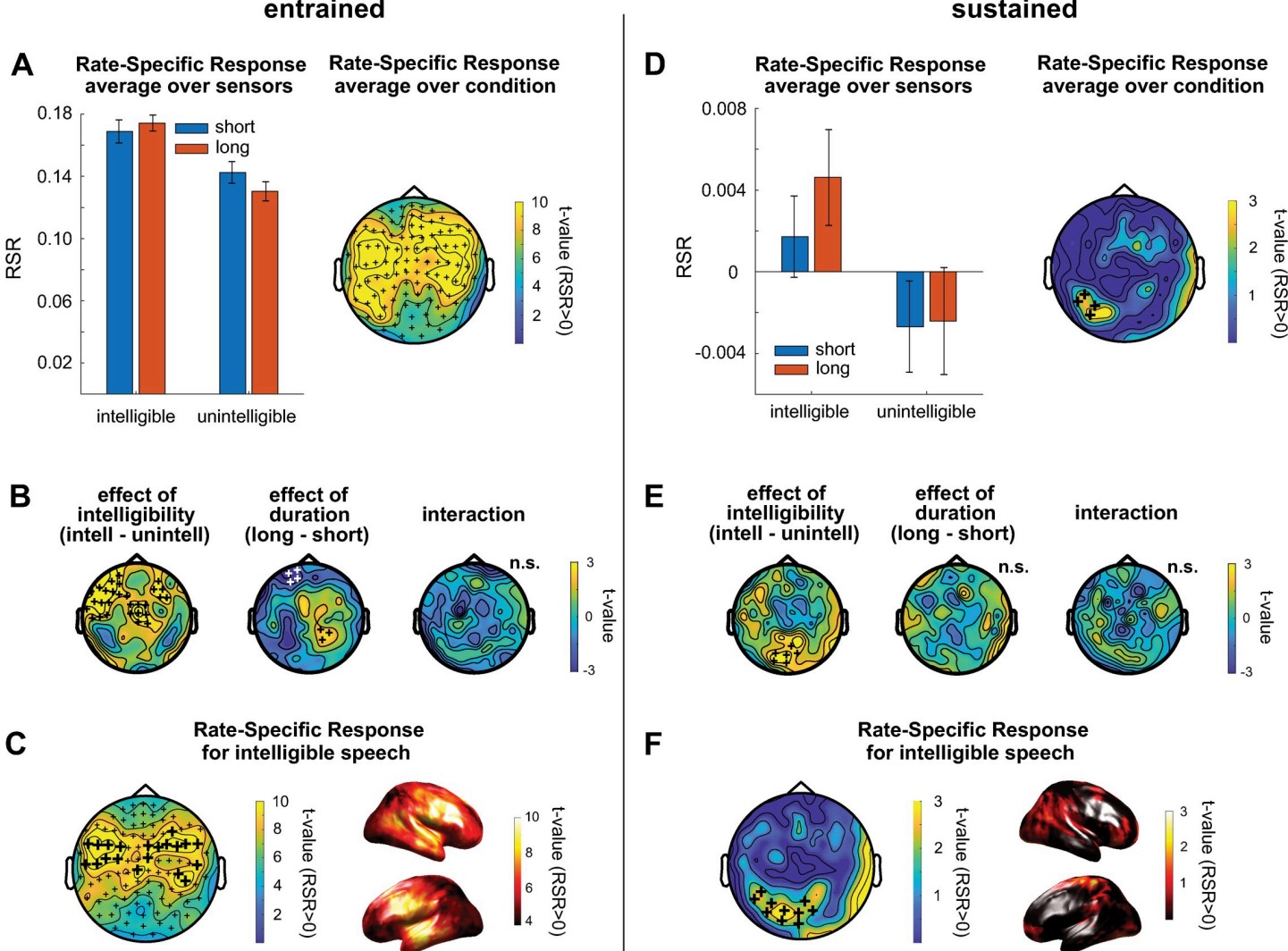

**Fig 2. Main results from Experiment 1.** (A–C) Results in the entrained time window. Bars in panel A show RSR in the different conditions, averaged across gradiometers and participants. Error bars show SEM, corrected for within-subject comparison. The topography shows t-values for the comparison with 0, separately for the 102 gradiometer pairs, and after RSR was averaged across conditions. Topographies in B contrast RSR across conditions. Topography and source plots in C show t-values for the comparison with 0 in the intelligible conditions. In all topographic plots, plus signs indicate the spatial extent of significant clusters from cluster-based permutation tests (see Materials and methods). In B, white plus signs indicate a cluster with negative polarity (i.e., negative t-values) for the respective contrast. In A and C, this cluster includes all gradiometers (small plus signs). In C, larger plus signs show the 20 sensors with the highest RSR, selected for subsequent analyses (Fig 3). (D–F) Same as A–C, but for the sustained time window. Please refer to S1 Data for the numerical values underlying this figure. RSR, rate-specific response; SEM, standard error of mean.

MEG sensors, the effect was localized to superior temporal regions and frontal regions bilaterally (Fig 2C).

In the sustained time window, when averaged across all conditions, the RSR was larger than 0 (cluster-based correction, $p = 0.05$; summed t = 9.22; 4 sensors in cluster) and maximal at left-lateralized parietal sensors (Fig 2D). When contrasting RSR across conditions (Fig 2E), we again found a main effect of intelligibility (cluster-based correction, $p = 0.01$; summed t = 15.84; 6 sensors in cluster), revealing stronger sustained RSRs for intelligible speech. Importantly, these sustained responses were only significant (i.e., RSR > 0) after intelligible speech (cluster-based correction, $p = 0.01$; summed t = 23.00; 10 sensors in cluster); no significant cluster was found after unintelligible speech. Sustained effects after intelligible

speech were localized to frontoparietal brain regions, with a peak in left parietal regions (Fig 2F).

To ensure that sustained oscillatory activity was not a result of aperiodic ("1/f") activity [23], which might differ between the 2 stimulus rates, we subtracted the "1/f component" from ITC measures of the sustained response (cf. [24]) by applying linear regression with reciprocal frequency (1/f) as a predictor of neural responses. We did this separately for the 2 stimulus rates, and recomputed the RSR using the residual (see Materials and methods). This analysis confirms a sustained oscillatory response only after intelligible speech (S1 Fig). Together, these effects demonstrate rhythmic brain responses at a frequency corresponding to the rate of stimulation, which outlast the stimulation at parietal sensors, and are present after intelligible, but not unintelligible rhythmic speech.

All sensors and conditions were included in our main analyses (Fig 2). We then explored the observed effects further (Fig 3), restricting analyses of orthogonal contrasts to sensors which are most important for those main results. For the entrained time window, we selected the 20 sensors with the largest RSR during intelligible speech (large plus signs in Fig 2C; the significant cluster included all sensors). For the sustained time window, we selected all 10 sensors in the significant cluster obtained after intelligible speech in (Fig 2F).

We first verified that the RSRs, revealed in our main analyses, were produced by responses at both of the stimulus rates tested. We found this to be the case in both entrained (Fig 3A) and sustained (Fig 3B) time windows: ITC at both 2 Hz and 3 Hz was significantly higher when it corresponded to the stimulation rate than when it did not (entrained: 2 Hz, $t(20) = 13.11$, $p < 0.0001$; 3 Hz, $t(20) = 11.46$, $p < 0.0001$; sustained: 2 Hz, $t(20) = 1.91$, $p = 0.035$; 3 Hz, $t(20) = 2.17$, $p = 0.02$). In the sustained time window, subtracting 1/f components (dashed lines in Fig 3B) from the data (continuous lines) revealed clearer peaks that correspond to the stimulation rate (or its harmonics). We note again the RSR discards such 1/f components by contrasting ITC values at the same 2 frequencies across the 2 stimulus rates.

We then tested how rhythmic responses developed over time. Both selected sensor groups (based on entrained and sustained responses) showed a significant RSR throughout the entrained time window (horizontal lines in Fig 3C; false discovery rate (FDR)-corrected). Importantly, the RSR at sensors selected to show a sustained response fluctuated at around the time of the first omitted word and then remained significantly above 0 during intelligible speech for most of the sustained time window. Although the presence of a sustained RSR is

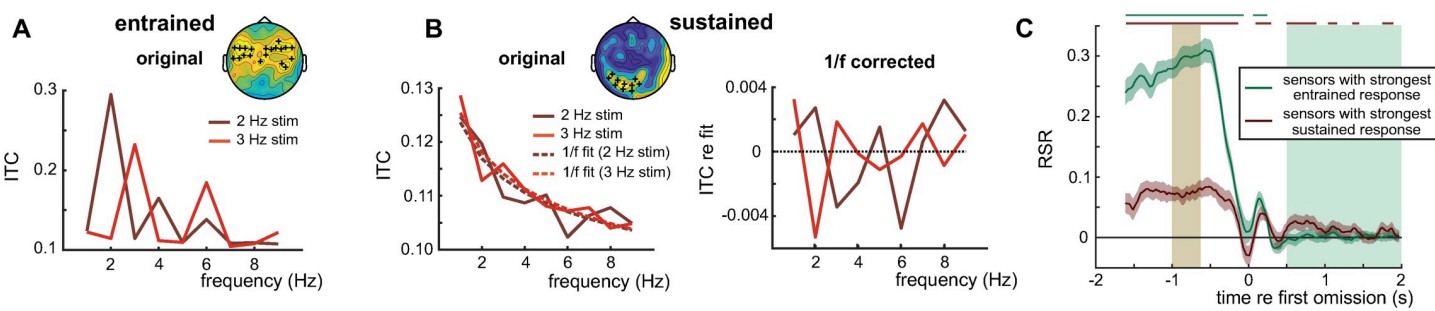

**Fig 3. Follow-up analyses from Experiment 1, using selected sensors (plus signs in insets, reproducing Fig 2C and 2F, respectively).** (A, B) ITC as a function of neural frequency, measured during (A) and after (B) intelligible speech, presented at 2 and 3 Hz. Note that these ITC values were combined to form RSR shown in Fig 2, as described in Fig 1F. For the right panel in B, a fitted "1/f" curve (shown as dashed lines in the left panel) has been subtracted from the data (see Materials and methods). Note that the peaks correspond closely to the respective stimulus rates, or their harmonics (potentially produced by imperfect sinusoidal signals). (C) RSR during intelligible speech as a function of time, for the average of selected sensors. Horizontal lines on top of the panel indicate an FDR-corrected $p$-value of $< = 0.05$ ($t$ test against 0) for the respective time point and sensor group. Shaded areas correspond to the 2 defined time windows (brown: entrained; green: sustained). Shaded areas around the curves show SEM. Please refer to S1 Data for the numerical values underlying this figure. FDR, false discovery rate; ITC, intertrial phase coherence; RSR, rate-specific response; SEM, standard error of mean.

expected (given the method used to select the sensors), this result gives us valuable insight into the timing of the observed effect. In particular, it excludes the possibility that the sustained effect is a short-lived consequence of the omission of an expected stimulus (see Discussion).

We did not measure the success of speech perception in Experiment 1. This is because such a task would have biased participants to attend differently to stimuli in intelligible conditions, making comparisons with neural responses in our unintelligible control condition difficult. Similarly, we refrained from using tasks which might have biased our measurement of endogenous oscillations in the silent period. For example, tasks in which participants are asked to explicitly predict an upcoming stimulus might have encouraged them to imagine or tap along with the rhythm. Our irregularity detection task was therefore primarily designed to ensure that participants remain alert and focused and not to provide behavioural relevance of our hypothesized sustained neural effect. Nevertheless, we correlated the RSR in both time windows (and at the selected sensors) with performance in the irregularity detection task (S2 Fig). We found a significant correlation between RSR in the entrained time window and detection performance (Pearson r = 0.53, $p$ = 0.01), demonstrating behavioural relevance of entrained brain responses. Perhaps unsurprisingly, given that there is no temporal overlap between the sustained response and target presentation, individual differences in the sustained RSR did not show a significant correlation with individual differences in rhythm perception (r = 0.27, $p$ = 0.28).

## Experiment 2: tACS produces sustained rhythmic fluctuations in word report accuracy

In Experiment 1, we showed sustained oscillatory activity after rhythmic sequences of intelligible speech, indicating that endogenous neural oscillations are involved in generating speech-entrained brain responses. In Experiment 2, we tested whether tACS produces sustained rhythmic changes in speech perception; if observed, this would not only provide an equivalent demonstration for tACS (i.e., that endogenous neural oscillations are entrained by transcranial electrical stimulation) but also show that these endogenous neural oscillations causally modulate perceptual outcomes.

Twenty participants were asked to report a single spoken, 16-channel vocoded target word, recorded rhythmically at 3 Hz, and embedded in background noise (Fig 4A). The signal–noise ratio between target word and noise was adjusted for individual participants, ensuring similar task difficulty across participants and ensuring that effects of tACS were not obscured by floor or ceiling report accuracy (see Materials and methods).

While participants performed this task, tACS was applied at 3 Hz over auditory regions, using the same configuration of bilateral circular and ring electrodes that yielded successful modulation of speech perception in [8] (see inset of Fig 4A). In each trial, the target word was presented so that its "perceptual centre" (see Materials and methods) falls at 1 of 6 different phase lags (red lines in Fig 4A), relative to tACS. Prior to target presentation, tACS was applied for approximately 3, 4, or 5 seconds. Importantly, the target word was presented either during tACS ("ongoing tACS"), which was turned off shortly afterwards, or immediately after tACS ("pretarget tACS"). We hypothesized that entrained neural activity due to tACS (irrespective of whether it involves endogenous oscillations; Fig 4B) will produce a phasic modulation of speech perception in the ongoing tACS condition, as reported previously [8–10]. However, in the pretarget tACS condition, such a phasic modulation can only be explained by sustained neural oscillations which lead to rhythmic changes in perception (Fig 4C).

Accuracy in reporting the target word was quantified using Levenshtein distance (similar to the proportion of phonemes reported correctly [25]; see Materials and methods). When

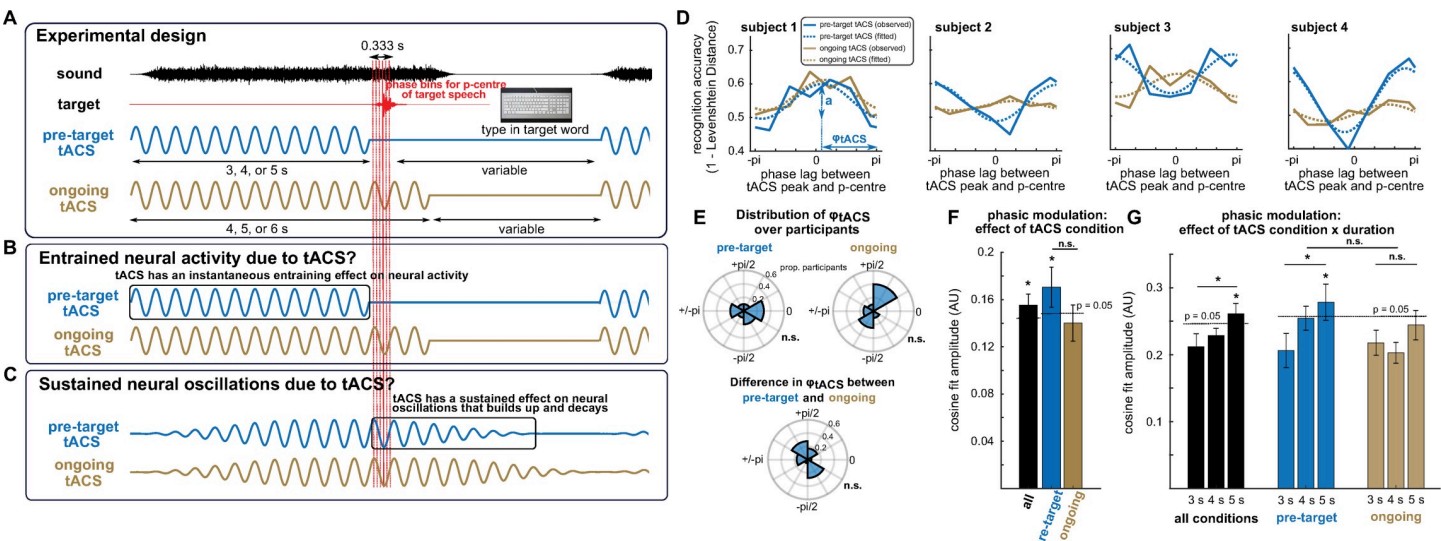

**Fig 4. Experimental paradigm and main results from Experiment 2.** (A) Experimental paradigm. In each trial, a target word (red), embedded in noise (black), was presented so that its p-centre falls at 1 of 6 different phase lags (vertical red lines; the thicker red line corresponds to the p-centre of the example target), relative to preceding ("pretarget tACS") or ongoing tACS (which was then turned off). After each trial, participants were asked to type in the word they had heard. The inset shows the electrode configuration used for tACS in both conditions. (B, C). Theoretical predictions. (B) In the case of entrained neural activity due to tACS, this would closely follow the applied current and hence modulate perception of the target word only in the ongoing tACS condition. (C) In the case that true oscillations are entrained by tACS, these would gradually decay after tACS offset, and a "rhythmic entrainment echo" might therefore be apparent as a sustained oscillatory effect on perception even in the pretarget condition. (D) Accuracy in the word report task as a function of phase lag (relative to tACS peak shown in (A), averaged across tACS durations, and for 4 example participants. Phasic modulation of word report was quantified by fitting a cosine function to data from individual participants (dashed lines). The amplitude (a) of this cosine reflects the magnitude of the hypothesized phasic modulation. The phase of this cosine ($\varphi_{tACS}$) reflects the distance between its peak and the maximal phase lag of π. Note that the phase lag with highest accuracy for the individual participants, estimated based on the cosine fit, therefore corresponds to π-$\varphi_{tACS}$. (E) Distribution of $\varphi_{tACS}$ in the 2 tACS conditions, and their difference. (F, G) Amplitudes of the fitted cosines (cf. amplitude a in panel D), averaged across participants. In (F), cosine functions were fitted to data averaged over tACS duration (cf. panel D). In (G), cosine functions were fitted separately for the 3 durations. For the black bars, cosine amplitudes were averaged across the 2 tACS conditions. Dashed lines show the threshold for statistical significance ($p < = 0.05$) for a phasic modulation of task accuracy, obtained from a surrogate distribution (see Materials and methods). Error bars show SEM (corrected for within-subject comparisons in (F)). Please refer to S1 Data for the numerical values underlying panels E–G. n.s., not significant; SEM, standard error of mean; tACS, transcranial alternating current stimulation.

averaged across phase lags, word report accuracy was slightly higher in the pretarget tACS condition ($0.50 \pm 0.09$, mean ± std) than in the ongoing tACS condition ($0.49 \pm 0.09$), but not significantly different ($t(19) = 1.67$, $p = 0.11$; repeated-measures $t$ test). This result indicates that the 2 tACS conditions did not reliably differ in their generic (i.e., phase-independent) effects on speech perception.

For each participant, and separately for the 2 tACS conditions, we determined how task accuracy varies with tACS phase lag (Fig 4D). We then fitted a cosine function to data from individual participants (dashed lines in Fig 4D). The amplitude of the cosine reflects how strongly speech perception is modulated by tACS phase. The phase of the cosine, labeled $\varphi_{tACS}$, reflects the distance between the peak of the cosine and the maximal phase lag tested (defined as π; Fig 4D). For example, a $\varphi_{tACS}$ of π would indicate highest word report accuracy at a tACS phase lag of 0.

Previous studies have reported that "preferred" tACS phase (leading to highest accuracy) varies across participants [7–10]. Indeed, in neither of the 2 conditions did we find evidence for a nonuniform distribution of $\varphi_{tACS}$ (Fig 4E) across participants (Rayleigh test for nonuniformity; pretarget tACS: $z(19) = 0.64$, $p = 0.53$; ongoing tACS: $z(19) = 0.71$, $p = 0.50$). We also failed to reveal a nonuniform distribution of the individual phase differences between conditions ($\varphi_{tACS(ongoing)}-\varphi_{tACS(pre-target)}$; $z(19) = 0.24$, $p = 0.79$), indicating that the perceptual outcome in the ongoing and pretarget tACS conditions might not rely on identical neural processes.

To statistically evaluate the hypothesized phasic modulation of word report accuracy, we compared the observed cosine amplitudes (Fig 4F and 4G) with a surrogate distribution—an approach which has recently been shown to be highly sensitive to detect such a phasic effect [21]. The surrogate distribution was obtained by repeatedly shuffling experimental variables assigned to individual trials and extracting cosine amplitudes for each of those permutations. Here, these variables can refer to tACS phase lags, conditions, or durations, depending on the comparison of interest (see Materials and methods).

We first pooled data over tACS durations (3, 4, and 5 seconds) before extracting cosine amplitudes (Fig 4F). When tACS conditions were combined (i.e., their cosine amplitudes averaged), we found a significant phasic modulation of word report accuracy ($z(19) = 2.80$, $p = 0.003$). When conditions were analysed separately, we found a significant phasic modulation of word report accuracy in the pretarget tACS condition ($z(19) = 2.96$, $p = 0.002$). This effect was not statistically reliable in the ongoing tACS condition ($z(19) = 0.98$, $p = 0.16$). However, the difference in modulation strength between tACS conditions was not significantly different from that obtained in a surrogate distribution ($z(19) = 1.37$, $p = 0.17$), indicating that the 2 conditions did not reliably differ in their efficacy of modulating speech perception.

We next tested whether the phasic modulation of speech perception depends on tACS duration (Fig 4G). When tACS conditions were combined, we found an increase in phasic modulation of word report accuracy from 3-second tACS to 5-second tACS that was significantly larger than that observed in a surrogate distribution ($z(19) = 1.82$, $p = 0.03$). After 5 seconds of tACS, the phasic modulation was significant ($z(19) = 2.36$, $p = 0.01$), while the modulation was not statistically reliable after 3 seconds of stimulation ($z(19) = -0.52$, $p = 0.70$). When tACS conditions were analysed separately, a significant effect of duration was observed in the pretarget tACS condition ($z(19) = 1.86$, $p = 0.03$), but not in the ongoing tACS condition ($z(19) = 0.69$, $p = 0.24$). After 5 seconds of tACS, the phasic modulation of word report accuracy was significant in the pretarget tACS condition ($z(19) = 2.15$, $p = 0.016$), but not in the ongoing tACS condition ($z(19) = 1.17$, $p = 0.12$). However, when effects of duration (3-second tACS versus 5-second tACS) were compared across tACS conditions, we did not find a reliable difference between the two ($z(19) = 0.90$, $p = 0.37$), indicating that there was no significant interaction between tACS condition and duration.

Together, we found rhythmic changes in speech perception after the offset of tACS, which depend on the duration of the preceding stimulation. This finding demonstrates that tACS can induce rhythmic changes in neural activity that build up over time and continue beyond the period of stimulation. Both of these effects are consistent with endogenous neural oscillations being entrained by tACS.

## Experiment 1 versus 2: Phase of speech-entrained EEG predicts tACS effects in single participants

In line with previous research [7–10], we found that participants differ in the tACS phase leading to more or less accurate perception, reflected by $\varphi_{tACS}$ (Fig 4E). Although adapting tACS protocols to individual participants has been suggested as a crucial step to increase effect sizes and advance the field [26–28], neural parameters that can predict these individual differences remain elusive. Here, we report an analysis of combined data from 18 participants who participated in both our experiments. Rather than the MEG data reported earlier, we analysed the concurrent EEG data collected during Experiment 1 and relate this to tACS effects observed in Experiment 2 in the same participants. This is because EEG is methodologically closer related to tACS than MEG: Both tACS and EEG, but not MEG, are similarly affected by distortions in current flow in the skull and other, nonneural tissues [29–32]. We therefore tested whether we can use EEG data to predict individual differences in $\varphi_{tACS}$.

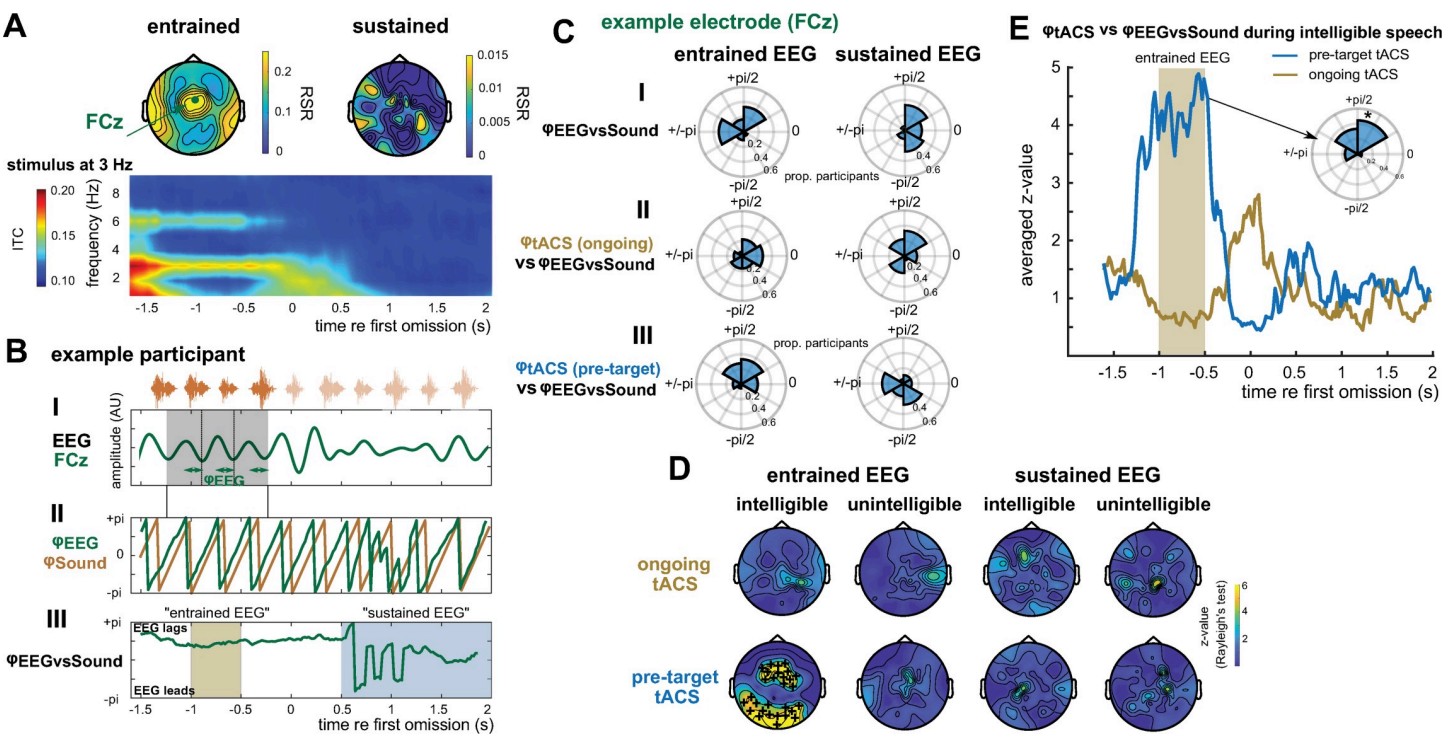

**Fig 5. Combining Experiments 1 and 2.** (A) EEG results from Experiment 1. Topographies show RSR in the intelligible conditions. The time–frequency representation depicts ITC during 3-Hz sequences, averaged across EEG electrodes, participants, and conditions (cf. Fig 1C). (B) Illustration of methodological approach, using example data from 1 participant and electrode (FCz, green in panel A). (B-I) Band-pass filtered (2–4 Hz) version of the EEG signal that has been used to estimate $\varphi_{EEG}$ in the panel below (B-II). In practice, EEG phase at 3 Hz was estimated using FFT applied to unfiltered EEG data. Consequently, $\varphi_{EEG}$ reflects the distance between the peaks of a cosine, fitted to data within the analysis window (shaded grey), and the end of each 3-Hz cycle (green arrows). (B-II) $\varphi_{EEG}$ (green; in the intelligible conditions and averaged across durations) and phase of the 3-Hz sequence ($\varphi_{Sound}$, orange). The latter is defined so that the perceptual centre of each word corresponds to phase π (see example sound sequence, and its theoretical continuation, on top of panel B-I). (B-III) Circular difference between $\varphi_{EEG}$ (green in B-II) and $\varphi_{Sound}$ (orange in B-II), yielding $\varphi_{EEGvsSound}$. Given that φ is defined based on a cosine, a positive difference means that EEG lags sound. (C) Distribution of individual $\varphi_{EEGvsSound}$, and its relation to $\varphi_{tACS}$. Data from 1 example electrode (FCz) is used to illustrate the procedure; main results and statistical outcomes are shown in panel D. (C-I) Distribution of $\varphi_{EEGvsSound}$ (cf. B-III), extracted in the intelligible conditions, and averaged across durations and within the respective time windows (shaded brown and blue in B-III, respectively). (C-II,III) Distribution of the circular difference between $\varphi_{tACS}$ (Fig 4E) and $\varphi_{EEGvsSound}$ (C-I). Note that a nonuniform distribution (tested in panel D) indicates a consistent lag between individual $\varphi_{tACS}$ and $\varphi_{EEGvsSound}$. (D) Z-values (obtained by means of a Rayleigh test; see Materials and methods), quantifying nonuniformity of the distributions shown in C-II,III for different combinations of experimental conditions. Plus signs show electrodes selected for follow-up analyses (FDR-corrected $p < = 0.05$). (E) Z-values shown in D for intelligible conditions as a function of time, averaged across selected EEG sensors (plus signs in D). For the electrode with the highest predictive value for tACS (F3), the inset shows the distribution of the circular difference between $\varphi_{tACS}$ and $\varphi_{EEGvsSound}$ in the pretarget condition, averaged within the entrained time window (shaded brown). Please refer to S1 Data for the numerical values underlying panels A, C–E. EEG, electroencephalography; FDR, false discovery rate; FFT, fast Fourier transformation; ITC, intertrial phase coherence; RSR, rate-specific response; tACS, transcranial alternating current stimulation.

In line with the MEG results reported earlier, EEG data in Experiment 1 showed a highly reliable RSR in the entrained time window (Fig 5A; $p < 0.001$; cluster-based correction). The RSR in the sustained time window was largest at frontoparietal electrodes, similar to our reported findings in MEG. However, this sustained effect was not statistically reliable (i.e., no significant clusters were obtained). This could either be due to the lower signal-to-noise ratio of EEG or because EEG and MEG measure nonidentical neural sources [33], which makes it possible that only 1 of the 2 methods captures a neural process of interest.

Although the RSR combines ITC measured during 2 different stimulus rates (Fig 1E and 1F), we here focused on EEG responses at 3 Hz in response to 3-Hz sequences, corresponding to the frequency of tACS in Experiment 2. Fig 5B and 5C illustrates our analysis procedure for 1 example participant (Fig 5B) and EEG electrode (Fig 5B and 5C). For each EEG electrode, we extracted the phase of the 3-Hz response at each time point throughout the trial and labeled

it $\varphi_{EEG}$ (Fig 5B-II, green). We used fast Fourier transformation (FFT) to estimate $\varphi_{EEG}$ (see Materials and methods), which is equivalent to fitting a cosine at the frequency of interest (i.e., 3 Hz) to data in the analysis window (shaded grey in Fig 5B-I) and extracting its phase. The value of $\varphi_{EEG}$ therefore corresponds to the distance between each of the 3 peaks of the fitted cosine and the end of the corresponding cycle (defined as π; Fig 5B-I).

For each participant and EEG electrode, we determined how $\varphi_{EEG}$ relates to the timing of the presented sound sequences ($\varphi_{Sound}$; Fig 5B-II, blue). Assuming rhythmic EEG responses reliably following the presented sequences, the phase relation between EEG and sound (i.e., their circular difference) should be approximately constant over time. This phase relation, labeled $\varphi_{EEGvsSound}$ (Fig 5B-III), was therefore averaged within each of the 2 time windows of interest (entrained and sustained). The distribution of $\varphi_{EEGvsSound}$ across participants in these time windows is shown in Fig 5C-I for the selected EEG electrode.

For each participant, EEG electrode, and the 2 time windows, we then calculated the (circular) difference between $\varphi_{EEGvsSound}$ and $\varphi_{tACS}$ in the ongoing (Fig 5C-II) and pretarget tACS conditions (Fig 5C-III), respectively. Importantly, a nonuniform distribution would indicate a consistent lag between $\varphi_{tACS}$ and $\varphi_{EEGvsSound}$ across participants. Fig 5D shows the degree of nonuniformity of these distributions (as the z-values obtained in Rayleigh test for nonuniformity; see Materials and methods), for all EEG electrodes, and different combinations of conditions in the 2 experiments. We found that the phase relation between EEG and intelligible speech in the entrained time window significantly predicts $\varphi_{tACS}$ in the pretarget tACS condition. This effect was maximal at frontocentral EEG electrodes (e.g., F3: z(17) = 8.88, $p$ = 0.003, FDR-corrected for 70 electrodes). While main results are shown for all electrodes and conditions (Fig 5D), we again restricted follow-up analyses to those which are most relevant, and based on orthogonal contrasts. Here, we found that $\varphi_{EEGvsSound}$ was most predictive for $\varphi_{tACS}$ around the presentation of the last word in the sequence (Fig 5E). At the sensor with the strongest effect (F3), we observed a shift of approximately 90 degrees (corresponding to approximately 83.3 ms) between $\varphi_{tACS}$ and $\varphi_{EEGvsSound}$ (inset in Fig 5E). As expected from its increased dissimilarity to tACS, MEG responses measured in Experiment 1 did not reveal any predictive value for tACS results from Experiment 2 (S3 Fig).

Findings shown in Fig 5 have important implications for future studies: Given the previous reports of tACS-induced changes in speech processing [7–11], tACS may be a promising tool to treat conditions associated with deficits in speech comprehension. However, individual differences in $\varphi_{tACS}$ have so far hampered this goal—existing data suggest that different tACS phases will lead to optimal perception for each individual participant, meaning that extensive testing might be needed to determine this optimal phase before further interventions. Based on the consistent phase shift between $\varphi_{EEGvsSound}$ and $\varphi_{tACS}$ shown in Fig 5E, however, it should be possible to predict optimal tACS phase for single participants from EEG responses aligned to rhythmic intelligible speech. We tested this prediction in an additional analysis, as illustrated in Fig 6 (see also Materials and methods). This analysis was designed to illustrate the implications of findings depicted in Fig 5D for future applications (e.g., when optimising tACS methods for use in interventions), rather than for providing new results. We selected EEG data from the entrained time window and the EEG electrode (F3) which was most predictive for effects of pretarget tACS (Fig 5D), and behavioural data from the same tACS condition. Such a selection is permitted as main results were already reported—without preselection—in Fig 5D. For each participant $i$, we determined their individual $\varphi_{EEGvsSound}$ (Fig 6B) and used it to estimate their individual $\varphi_{tACS}$ (Fig 6C), based on the difference between the two that was observed on the group level (Fig 6A and 6C). Importantly, for the latter, data from participant $i$ were excluded, avoiding circularity of the procedure. For each participant, the estimated $\varphi_{tACS}$ was then used to predict the tACS phase lag with highest accuracy in the word report

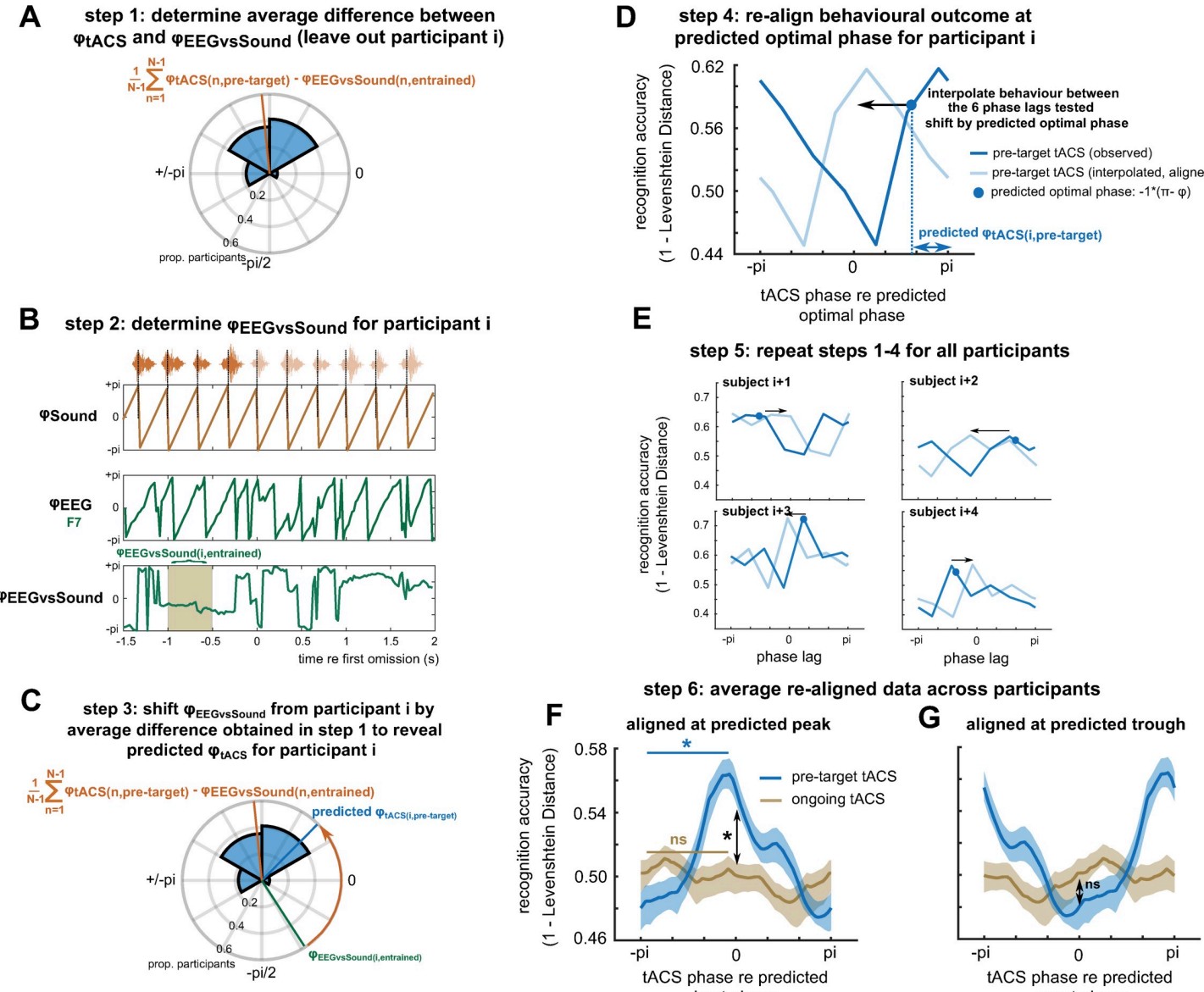

**Fig 6. Predicted individual preferred tACS phases in the pretarget tACS condition from EEG data measured in the entrained time window at sensor F3.** (A) Step 1: For each participant $i$, data from all remaining participants were used to estimate the average difference between $\varphi_{tACS}$ and $\varphi_{EEGvsSound}$. (B) Step 2: $\varphi_{EEGvsSound}$ was determined for participant $i$. (C) Step 3: This $\varphi_{EEGvsSound}$ was shifted by the phase difference obtained in step 1, yielding the predicted $\varphi_{tACS}$ for participant $i$. (D) Step 4: The predicted $\varphi_{tACS}$ was used to estimate the tACS phase lag with highest perceptual accuracy for participant $i$, and the corresponding behavioural data were shifted so that highest accuracy was located at a centre phase bin. Prior to this step, the behavioural data measured at the 6 different phase lags were interpolated to enable realignment with higher precision. (E) Step 5: This procedure was repeated for all participants. (F) Step 6: The realigned data were averaged across participants (blue). For comparison, the procedure was repeated for the ongoing tACS condition (using EEG data from the same sensor; brown). The shaded areas show SEM, corrected for within-subject comparison. (G). Same as in (F), but aligned at the predicted worst phase for word report accuracy. Please refer to S1 Data for the numerical values underlying panels F and G. EEG, electroencephalography; SEM, standard error of mean; tACS, transcranial alternating current stimulation.

task (blue dot in Fig 6D and 6E). The behavioural data collected in Experiment 2 were realigned, relative to this predicted optimal phase lag (Fig 6D; see S4 Fig for individual realigned data from all participants). The outcome, averaged across participants, is shown in Fig 6F (blue). As intended, word report accuracy was highest at the predicted optimal phase lag (0 in Fig 6F), and significantly higher than in the opposite phase bin (+/−π in Fig 6F),

which should lead to worst performance (t(17) = 4.49, $p < 0.001$). This result confirms that optimal tACS phases for speech perception can be estimated, exclusively based on individual EEG data (if the average difference between $\varphi_{tACS}$ and $\varphi_{EEGvsSound}$ is known).

## Sustained oscillations produced by tACS enhance, but do not disrupt speech perception

It remains debated whether a phasic modulation of speech perception, produced by tACS, reflects an enhancement or disruption of perception, or both [8–11,34]. Given that $\varphi_{EEGvsSound}$ was not predictive of $\varphi_{tACS}$ in the ongoing tACS condition (Fig 5D), we used data from the latter to test this question. We used the procedure illustrated in Fig 6 (using data from the same EEG sensor F3) to predict optimal tACS phases in the ongoing tACS condition (see Materials and methods). As $\varphi_{EEGvsSound}$ does not predict $\varphi_{tACS}$ in this condition, any tACS-dependent modulation of task accuracy should be abolished by the realignment, and the realigned data (Fig 6F, brown) should therefore reflect the null hypothesis, i.e., task outcome in the absence of a phasic modulation. Indeed, word report accuracy was not higher at the predicted optimal phase lag for the ongoing tACS condition than at the opposite phase lag (t(17) = 0.08, $p = 0.53$).

Given that entrained EEG is predictive for $\varphi_{tACS}$ only in the pretarget tACS condition (Fig 5D), there must be some phase bins in which accuracy differs between the 2 tACS conditions after EEG-based realignment. However, these previous analyses did not reveal the direction of this difference (enhancement versus disruption). We therefore compared performance at the predicted optimal tACS phase between the 2 tACS conditions and found higher word report accuracy in the pretarget tACS condition (t(17) = 3.48, $p = 0.001$). For both conditions, we then realigned the behavioural data again, but this time at the tACS predicted to be worst for performance (i.e., 180° away from the tACS phase predicted to be optimal for performance). Performance at the predicted worst tACS phase did not significantly differ between the 2 conditions (t(17) = 1.34, $p = 0.90$). These results show that the sustained phasic modulation of word report accuracy, produced by pretarget tACS, reflects an enhancement of speech perception both relative to a nonoptimal tACS phase and compared to EEG-aligned data from an ongoing tACS condition in which EEG data were not predictive of optimal tACS phase.

## Discussion

In 1949, Walter and Walter [35] observed that rhythmic sensory stimulation produces rhythmic brain responses. Importantly, in their paper, when listing potential explanations for their observation, they distinguished "fusion of evoked responses giving an accidental appearance of rhythmicity" from "true augmentation or driving of local rhythms at the frequency of the stimulus." Now, more than 70 years later, it remains an ongoing debate whether "neural entrainment," brain responses aligned to rhythmic input, is due to the operation of endogenous neural oscillations or reflects a regular repetition of stimulus-evoked responses [16,36–39]. In 2 experiments, we provide clear evidence for entrained endogenous neural oscillations, by showing that rhythmic brain responses and rhythmic modulation of perceptual outcomes can outlast rhythmic sensory and electrical stimulation. We will discuss the implication of these sustained effects of sensory and electrical stimulation, before considering the functional interpretation of neural after-effects. We finish by discussing the potential for practical application of our combined EEG and tACS findings in supporting impaired speech perception.

## Endogenous neural oscillations entrained by rhythmic sensory and electrical stimulation

Previous studies in a range of domains have similarly demonstrated sustained oscillatory effects after rhythmic sensory stimulation (summarized in [16]). Both perception and electrophysiological signals have been shown to briefly oscillate after a rhythmic sequence of simple visual [40–42] or auditory [43–45] stimuli, such as flashes or pure tones. A recent study showed that such a sustained rhythmic response occurs when preceded by a stimulus evoking the perception of a regular beat, but not when participants merely expect the occurrence of a rhythmic event [46]. Although neural entrainment is widely explored in speech research [1,2], we are only aware of 1 study reporting sustained oscillatory effects produced by human speech: Kösem and colleagues [17] showed that, immediately after a change in speech rate, oscillatory MEG responses can still be measured at a frequency corresponding to the preceding speech (summarized in [15]). Our results in Experiment 1 are in line with this study and extend it by showing that (1) sustained oscillations produced by speech can be measured in silence and (2) are not observed for acoustically matched speech stimuli that are unintelligible. Similar effects of intelligibility on neural entrainment have been described for combined tACS and fMRI: Neural responses in the STG to intelligible speech, but not to unintelligible speech, were modulated by tACS [7]. In Experiment 1, we also replicated our previous MEG finding of more reliable stimulus-aligned responses to intelligible than unintelligible speech [5,6]. We further show that (1) rhythmic responses to intelligible speech persist after the offset of the speech stimulus and that (2) this sustained effect is absent for acoustically matched, unintelligible speech. Our results should not be taken as evidence that endogenous neural oscillations are irrelevant for the processing of sounds other than human speech (e.g., [43–45]). However, they might suggest that endogenous oscillations are optimized to process speech, due to its quasi-rhythmic properties [3,47]. Additionally, it is possible that the increased salience of intelligible speech (as compared to noise or tone stimuli) enhances participants' alertness and encourages higher-level processing, which has been shown to lead to enhanced oscillatory tracking of rhythmic structures [48,49]. Together, our MEG findings suggest that endogenous neural oscillations are active during neural entrainment and that these oscillatory mechanisms are of particular importance for processing intelligible speech.

It is well established that the omission of an expected stimulus evokes a prominent neural response [50–53]. One concern that could be raised regarding the present findings is whether our sustained effects could have been generated by an omission response rather than true oscillatory activity. Several aspects of our Experiment 1 suggest that omission-evoked responses are unlikely to explain the sustained effects of rhythmic stimulation: (1) omission responses would only lead to a sustained RSR if they were specific to the stimulation rate (i.e., if the omission leads to an increase in 2-Hz ITC after 2-Hz sequences and 3-Hz ITC after 3-Hz sequences); (2) sustained oscillatory activity after the end of a sequence lasts longer than would be expected from a single, punctate omission response (see Fig 3C); (3) previous observations of omission responses show that these are largely generated in brain regions that were most active while rhythmic stimuli were presented [52,53], whereas our study showed sustained responses in brain regions that were not the primary driver of responses measured during sensory stimulation (compare scalp topographies and source distributions in Fig 2C and 2F). These findings therefore suggest that sustained activity is generated by true oscillatory neural activity produced in response to intelligible speech.

Several studies have reported modulation of speech perception outcomes by tACS and conclude that changes in neural entrainment, produced by varying the phase relation between tACS and speech rhythm, are responsible [8–11]. However, thus far, these effects could reflect

the rhythmic nature of the applied current, which might interfere with processing of speech presented with the same rhythm without any involvement of neural oscillations [15]. In Experiment 2, we found sustained rhythmic fluctuations in speech perception that continued after the offset of tACS. Our results are an important extension of previous work as they suggest that (1) modulation of speech perception can be due to the operation of neural oscillations entrained by tACS and that (2) sustained oscillatory effects after tACS can be measured in word report outcomes and hence are causally relevant for speech perception. These findings for speech have precedent in other sensory modalities and brain regions. For example, a recent study [54] used tACS at 7 Hz to stimulate parietal-occipital regions and reported sustained rhythmic EEG responses at the frequency of electric stimulation. Although the functional role of these sustained neural effects for perceptual processes (such as perceptual integration) remain unclear, this previous study provides evidence for neural oscillations entrained by tACS that parallels the present work. The tACS method used here, in which perceptual effects are observed subsequent to the end of electrical stimulation, are clearly amenable to further exploration in studies combining tACS and EEG.

In Experiment 2, the phasic modulation of speech perception observed after tACS (in the pretarget tACS condition) was not significantly different from that during tACS (in the ongoing tACS condition). In light of results from Experiment 1, where the sustained rhythmic response was clearly weaker than the entrained one, this might seem surprising. Importantly, however, the process that interferes with our ability to measure endogenous oscillations during rhythmic stimulation is not identical in the 2 experiments. In Experiment 1, rhythmic sensory stimulation produced strong, regular evoked activity which dominates the response in the entrained time window. In Experiment 2, the current applied during tACS alternated regularly between periods of strong stimulation (at the tACS peaks and troughs) and no stimulation (at the zero crossings). This, according to our assumptions, might produce rhythmic modulation of speech perception that does not necessarily involve endogenous oscillations (perception might simply "follow" the amount of current injected). However, tACS is not strong enough to evoke neural activity [55,56], and the described effect will not dominate responses as strongly as sensory stimulation in Experiment 1. Moreover, such a phasic effect on speech perception does not necessarily combine additively with that produced by entrained endogenous oscillations—indeed, these 2 processes might even interfere with each other. Consequently, and in line with our results, rhythmic modulation of speech perception is not necessarily expected to be stronger when both processes interact (regular changes in current versus entrained oscillations in the ongoing tACS condition) as compared to an effect that is due to endogenous oscillations alone (in the pretarget tACS condition).

Another line of evidence for endogenous oscillations entrained by a rhythmic stimulus comes from studies testing how brain responses vary as a function of stimulus rate and intensity (summarized in [16]). It is a clear prediction from classical physical models that the intensity required to entrain endogenous oscillations decreases when the rate of the entraining stimulus approaches their natural frequency [57–60]. Indeed, this phenomenon, termed "Arnold Tongue," has recently been observed for visual stimulation [61]. There is tentative evidence that tACS-induced responses behave in a similar way (summarized in [59]), but more studies are needed to substantiate this claim. Based on similar reasoning, entrainment effects should also be stronger when the system has "more time" to align with the external oscillator [59,62]. Our finding that tACS effects on perception increase with stimulation duration (Fig 4G) is therefore clearly in line with oscillatory models. Importantly, such a behaviour was apparent in the pretarget tACS condition, in which effects of endogenous oscillations could be distinguished from those of other, potentially interfering neural processes. Although effects of tACS duration on behaviour were numerically larger and only statistically reliable in this

condition, we hesitate to conclude that the effect is specific to pretarget tACS since the condition by duration interaction was not reliable. Nevertheless, this result not only adds to existing demonstrations of endogenous oscillations entrained by tACS but also points to entrained neural oscillations being more than just a passive response to rhythmic input. This idea is discussed in detail in the next section.

## Rhythmic entrainment echoes—Active predictions or passive after-effect?

In both our MEG and tACS experiments, we demonstrate that entrained neural and perceptual processes are more than a simple reflection of rhythmic input driving an otherwise silent system (Fig 7A): Based on the observation of sustained oscillatory responses after stimulus offset, we conclude that an endogenous oscillatory system is involved in such entrained brain

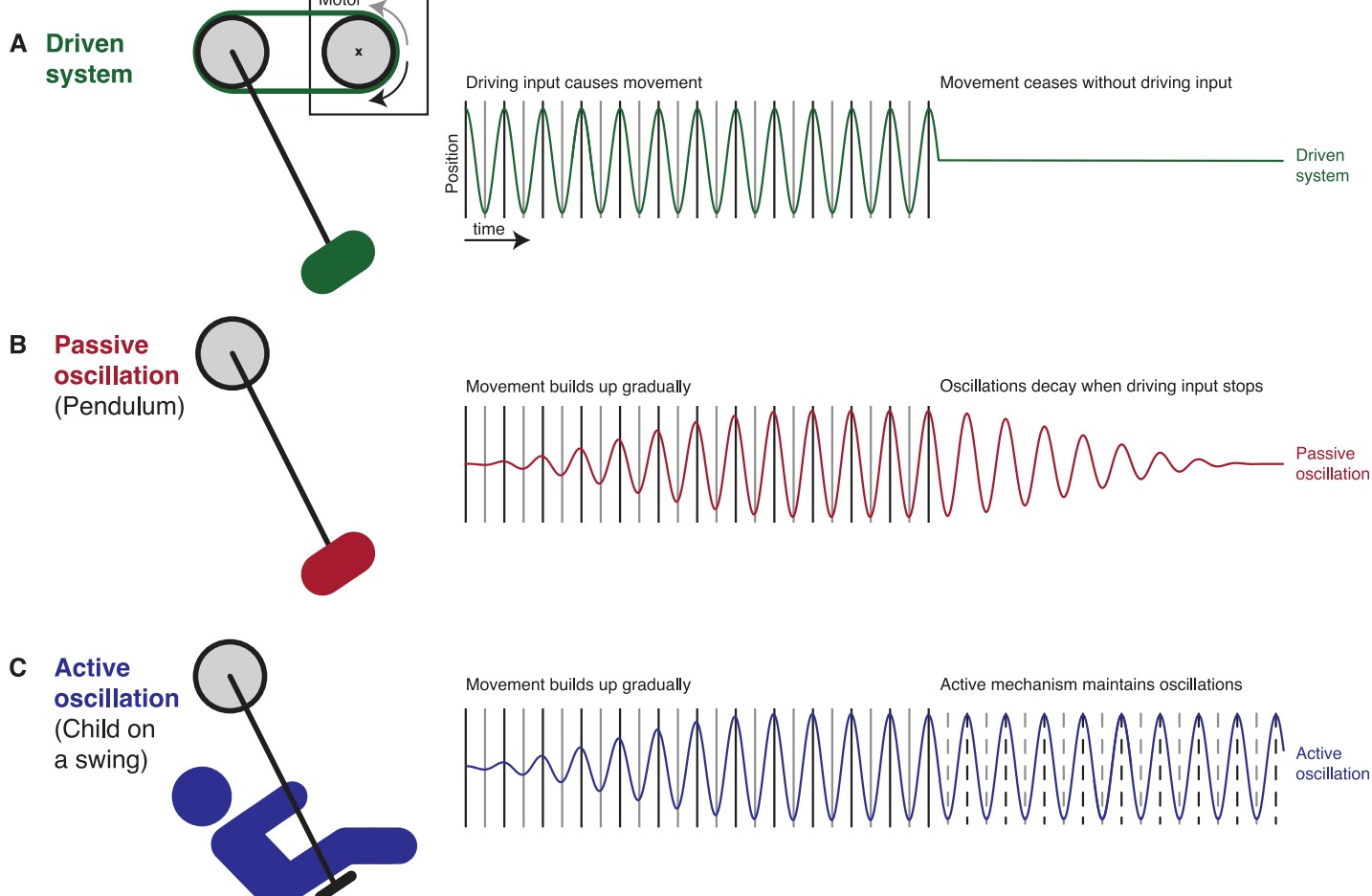

**Fig 7. Three physical models that could be invoked to explain neural entrainment, and their potential to explain rhythmic entrainment echoes.** (A) In a system without any endogenous processes (e.g., neural oscillations), driving input would produce activity which ceases immediately when this input stops. (B) A more direct account of rhythmic entrainment echoes is that endogenous neural oscillations resemble the operation of a pendulum which will start swinging passively when "pushed" by a rhythmic stimulus. When this stimulus stops, the oscillation will persist but decays over time, depending on certain "hard-wired" properties (similar to the frictional force and air resistance that slows the movement of a pendulum over time). (C) Endogenous neural oscillations could include an active (e.g., predictive) component that controls a more passive process—similar to a child that can control the movement of a swing. This model predicts that oscillations are upheld after stimulus offset as long as the timing of important upcoming input (dashed lines) can be predicted. Note that, for the sake of clarity, we made extreme predictions to illustrate the different models. For instance, depending on the driving force of the rhythmic input, pendulum and swing could reach their maximum amplitude near-instantaneously in panels B and C, respectively, and therefore initially resemble the purely driven system shown in A. Similarly, it is possible that the predictive process (illustrated in C) operates less efficiently in the absence of driving input and therefore shows a decay similar to that shown by the more passive process (shown in B).

responses. Although endogenous oscillations are difficult to measure during stimulation, the most parsimonious explanation of our results is that the entrained response entails both evoked responses and endogenous oscillations, with the former dominating the response. After stimulus offset, only the latter prevails, leading to a change in topographical pattern and estimated source. Indeed, we found that sensors capturing sustained oscillations also show a significantly entrained response during sensory stimulation (Fig 3C, red), while stronger, stimulus-driven activity at distinct sensors quickly subsided after stimulation (green in Fig 3C).

What is the neural mechanism and functional role played by these rhythmic echoes of previously entrained responses (hereafter, "entrainment echoes," cf. [54])? We here illustrate 2 different, but not mutually exclusive, models which can explain the observed entrainment echoes. In one model, these rhythmic echoes reflect the passive reverberation of an endogenous neural oscillation that has previously been activated by a rhythmic stimulus. A physical analogy for this would be a pendulum that responds to a regular "push" by swinging back and forth and that continues to produce a regular cyclical movement without external input until its kinetic energy has subsided (Fig 7B). In the other model, stimulus-aligned oscillations are the result of an active mechanism that, through predictive processes, comes to align the optimal (high-excitability) oscillatory phase to the expected timing of important sensory or neural events [12,13]. In this view, oscillatory activity can be actively maintained after stimulus offset and can persist for as long as these predictions are required. It is plausible that this active component is imposed onto a more "hard-wired," passive mechanism, that is oscillations might be entrained passively, but that this mechanism is under top-down control and can be adjusted if necessary. A physical analogy for this is the way in which a child will move on a swing if pushed, but can also control whether or not the movement of the swing is sustained after their helper stops pushing (Fig 7C). The active mechanism, in this case, is the timing and amplitude of small movements that a sufficiently skilled child can coordinate with the movement of the swing to maintain oscillations without external help.

Several of our observations do point to an "active" component involved in generating rhythmic entrainment echoes; however, providing a definitive answer to this question remains for future studies. In both experiments, we found that the neural systems involved in producing sustained effects are distinct from those that are most active during the presence of the rhythmic stimulus. In Experiment 1, sustained MEG oscillations were maximal at parietal sensors and had a clearly different scalp topography and source configuration from typical auditory responses (cf. [46] for a similar shift towards parietal sensors after rhythmic stimulation). In Experiment 2, individual tACS phase lags leading to highest word report accuracy after tACS offset were unrelated to those measured during tACS. Together, these findings are important as they speak against purely "bottom-up" or stimulus-driven generators of sustained oscillatory responses that merely continue to reverberate for some time after stimulus offset. Instead, they suggest that a distinct oscillatory network seems to be involved that might be specialized in "tracking" and anticipating important upcoming sensory events—potentially by adjusting and modulating a more passive, sensory processing system that aligns to rhythmic speech stimuli. It is possible that we can mimic such top-down effects using tACS, providing rhythmic predictions to auditory regions using electrical stimulation.

This proposal that top-down predictions for the timing of upcoming stimuli are achieved using neural oscillations is also in line with previous studies suggesting that neural predictions are fundamental for how human speech is processed by the brain [25,63–65]. It is possible that predictive oscillatory mechanisms are particularly strong for intelligible speech, and therefore upheld for some time when the speech stops. In contrast, unintelligible noise-like sequences, typically irrelevant in everyday situations, might lead to weaker predictions or shorter-duration sustained responses—explaining the results observed in Experiment 1.

Stronger rhythmic responses during intelligible than unintelligible speech [5,6], as well as sustained oscillatory effects for speech sounds [17], have previously been shown in auditory brain areas. However, all of these studies measured neural effects during auditory input, which might bias localization of the neural responses towards auditory areas. Our study, in contrast, revealed sustained effects during poststimulus silent periods at parietal sensors. This method might therefore yield a more precise estimate of where these effects originate. Auditory input fluctuates rapidly, which requires the auditory system to quickly adapt its oscillations to changes in input [66,67]. Auditory input is represented more faithfully (i.e., less abstractly), and therefore on a faster time scale, in auditory brain regions than in "higher-level" ones [68]. Thus, it is possible that oscillatory activity in the former involves more immediate responses and hence disappears quickly after sound offset. In contrast, a more abstract representation of a rhythmic input—including phasic predictions about timing—might be more stable over time and can remain present even after stimulus offset. This might be another reason to explain why our sustained oscillatory effects were found to be maximal at parietal sensors, potentially reflecting neural activity at a higher level of the cortical hierarchy.

## Predicting tACS outcomes from EEG data—Implications for future work and applications

It is a common observation that participants differ in how they respond to a given tACS protocol. For example, there is typically no consistent tACS phase which leads to highest perceptual accuracy for all participants [7–10]. Individualizing brain stimulation protocols has therefore been proposed as a crucial step to advance the theoretical and practical application of this line of research [26–28]. A recent study [69] reported that the phase relation between tACS and visual flicker modulates the magnitude of EEG responses to the flicker when tACS is turned off. Moreover, the individual "best" phase relation between tACS and flicker (leading to strongest EEG responses) was correlated with the individual phase relation between EEG and flicker. We replicate and extend this finding in a new modality by showing that the individual phase lag between EEG and intelligible speech can predict which tACS phase leads to more or less accurate perception in the same participant. Indeed, we found that EEG data from individual participants is sufficient to predict which tACS phase is optimal for perception, so long as the average lag between the two can be estimated even when using other, independent participants (Fig 6). This result is important, as it shows that tACS can be adapted to individual brains based on EEG observations and establishes a method for aligning EEG and tACS findings for single participants. In an applied setting, these methods make the application of brain stimulation more efficient since the search for the most effective phase can be guided by EEG data rather than by trial and error. This finding therefore increases the potential for clinical or educational applications of tACS methods in future.

Perhaps surprisingly, given results from Experiment 1, the phase of the entrained, but not sustained EEG response, was predictive for the phase of the sustained tACS effect. This result might be explained by the fact that, possibly due the lower signal to noise ratio of EEG, the sustained oscillatory response was not statistically reliable in the EEG in Experiment 1 (Fig 5A). Consequently, a link between sustained oscillatory effects in EEG and tACS might not have been detectable, even if it exists, simply because the former was not measured reliably. Nevertheless, our finding that the entrained EEG response predicts sustained tACS phase indicates that entrained EEG responses can capture the phase of endogenous oscillations, despite observations of simultaneous evoked neural activity. MEG, showing statistically robust sustained responses (Fig 2), is not as closely related to tACS as EEG (as its signal is not affected by the same distortions by bone and tissue) and is therefore less likely to be predictive of tACS

outcomes (cf. S3 Fig). Future studies may need electrophysiological methods with higher signal-to-noise ratio than EEG, such as electrocorticography, ECoG, to test the relationship between sustained neural responses and tACS-induced changes in perception in more detail.

According to the simplest interpretation of the reciprocity between EEG and tACS, if the signal from a neural source is captured at a certain (EEG) electrode position, then the same electrode position should be efficient in stimulating this neural source (with tACS) [30–32]. Vice versa, if a tACS electrode configuration is successful in targeting a certain neural source, then activity from this source should be measurable with EEG at this electrode position. As the topographical pattern of EEG signals with high predictive value for tACS (fronto-occipital pattern; Fig 5D) was different from the tACS electrode position (T7/8), our results indicate that this simple interpretation does not hold and that more complex mechanisms underlie our observations. This could be because multiple neural sources are involved and interact to produce the topographical distribution measured with EEG, while the tACS protocol used can only reach one or some of them. It is also possible that tACS modulates the efficacy of sensory input to activate neural ensembles, while EEG measures the output of these ensembles. Differences in neural populations contributing to input versus output processing, including their orientation to the scalp, might explain the observed deviance from simple reciprocity between EEG and tACS. Finally, it is possible that even stronger modulation of perception could be achieved if tACS were applied at those (fronto-occipital) EEG electrode positions showing maximal predictive values for tACS effects—this could be explored in future work.

It is of note that the phasic modulation of speech perception was not statistically reliable when the target was presented during tACS (i.e., in the ongoing tACS condition). This result seems in contrast to previous work [7–11]. However, in those studies, participants listened to and reported longer speech sequences while they were asked to detect a single target word (presented in background noise) in the current study. The quasi-regular rhythm of such sequences might act as an additional entraining stimulus which could boost or interact with tACS effects (see also next paragraph), in particular when perception is tested during tACS. Future studies should test the interesting question of whether and how the rhythmicity of the speech stimulus affects the efficacy of tACS during and after its application.

In previous work, using the same electrode configuration as applied in Experiment 2, we reported that tACS can only disrupt, and not enhance speech perception [8]. We previously hypothesized that this is because tACS was applied simultaneously with rhythmic speech sequences, which as Experiment 1 of our study shown can themselves entrain brain activity. If neural entrainment to the speech sequences were already at the limit of what is physiologically possible, tACS might only be able to disrupt, but not to enhance it further. Importantly, in the current study, tACS was applied during nonrhythmic background noise, i.e., without any simultaneously entraining auditory stimulus. Our finding of enhanced speech perception therefore supports the hypothesis that tACS can enhance neural entrainment. However, if it is applied simultaneously with a strong "competing" entraining stimulus, tACS might only be able to disrupt entrainment. Together with the finding that tACS can be individualized, the protocol used here seems a promising method for future technological applications in which tACS is used to enhance speech perception in a real-world setting.

In conclusion, we report evidence that endogenous neural oscillations are a critical component of brain responses that are aligned to intelligible speech sounds. This is a fundamental assumption in current models of speech processing [1] that we believe is only now clearly established by empirical evidence. We further show that tACS can modulate speech perception by entraining endogenous oscillatory activity. In this way, we believe our work critically advances our understanding of how neural oscillations contribute to the processing of speech in the human brain.

## Materials and methods

### Participants

A total of 24 participants were tested after giving written informed consent in a procedure approved by the Cambridge Psychology Research Ethics Committee (application number PRE.2015.132) and carried out in accordance with the Declaration of Helsinki. Three participants did not finish Experiment 1, leaving data from 21 participants (10 females; mean ± SD, 37 ± 16 years) for further analyses; 4 participants did not finish Experiment 2, leaving 20 participants for further analyses (11 females; 39 ± 15 years). Eighteen participants (9 females; 40 ± 15 years) finished both experiments.

All participants were native English speakers, had no history of hearing impairment, neurological disease, or any other exclusion criteria for MEG or tACS based on self-report.

### Stimuli

Our stimuli consisted of a pool of approximately 650 monosyllabic words, spoken to a metronome beat at 1.6 Hz (inaudible to participants) by a male native speaker of British English (author MHD). These were time-compressed to 2 and 3 Hz, respectively, using the pitch-synchronous overlap and add (PSOLA) algorithm implemented in the Praat software package (version 6.12). This approach ensures that "perceptual centres," or "p-centres" [70] of the words were aligned to the metronome beat (see vertical lines in Fig 1C) and, consequently, to rhythmic speech (in perceptual terms). Moreover, the well-defined rhythmicity of the stimulus allows a precise definition of the phase relation between stimulus and tACS (see below).

For Experiment 1 (Fig 1A), these words were combined to form rhythmic sequences, which were 2 or 3 seconds long and presented at 1 of 2 different rates (2 or 3 Hz). Depending on the duration and rate of the sequence, these sequences therefore consisted of 4 (2 Hz/2 seconds), 6 (3 Hz/2 seconds and 2 Hz/3 seconds) or 9 words (3 Hz/3 seconds). Noise-vocoding [18] is a well-established method to produce degraded speech which varies in intelligibility, depending on the number of spectral channels used for vocoding. In Experiment 1, we used highly intelligible 16-channel vocoded speech and 1-channel noise-vocoded speech, which is a completely unintelligible, amplitude-modulated noise (for more details, see [7,8]). Importantly, noise-vocoding does not alter the rhythmic fluctuations in sound amplitude of the stimulus that are commonly assumed to be important for neural entrainment [47]. Thus, acoustic differences in the broadband envelope between the 2 conditions cannot be responsible for differences in the observed neural responses.

For Experiment 2 (Fig 4A), we presented participants with single 16-channel noise-vocoded target words, time-compressed to 3 Hz. These words were embedded in continuous noise with an average spectrum derived from all possible (approximately 650) target words. The noise was presented for approximately 5 to 7 seconds. The target word occurred between 2 and 1.722 second before noise offset, depending on its phase lag relative to tACS (see Experimental design and Fig 4A). The noise was faded in and out at the beginning and end of each trial, respectively. All stimuli were presented to participants via headphones (through insert earphones connected via tubing to a pair of magnetically shielded drivers in Experiment 1; ER-2 insert earphones in Experiment 2; Etymotic Research, United States of America).

### Experimental design

In Experiment 1, while MEG/EEG data were recorded, participants listened to the rhythmic sequences (Fig 1A) and pressed a button as soon as they detected an irregularity in the sequence rhythm (red in Fig 1A). The irregularity was present in 12.5% of the sequences and

was produced by shifting one of the words (excluding first and last) in the sequence by ± 68 ms. Participants completed 10 experimental blocks of 64 trials each. For each block, the rate of the sequences was chosen pseudorandomly and kept constant throughout the block. In each trial, the intelligibility (16- or 1-channel speech) and duration (2 or 3 seconds) of the sequence was chosen pseudorandomly. Consequently, participants completed a total of 80 trials for each combination of conditions (rate × intelligibility × duration). Each of the sequences was followed by a silent interval in which sustained oscillatory responses were measured (Fig 1C). These silent intervals were $2 + x$ s long, where $x$ corresponds to 1.5, 2, or 2.5 times the period of the sequence rate (i.e., 0.75, 1, or 1.25 second in 2-Hz blocks, and 0.5, 0.666, or 0.833 second in 3-Hz blocks). $x$ was set to 2 in 50% of the trials.

In Experiment 2, tACS was applied at 3 Hz, and participants were asked to identify a target word embedded in noise and report it after each trial using a standard computer keyboard. The start and end of each trial was signaled to participants as the fade in and out of the background noise, respectively (Fig 4A). The next trial began when participants confirmed their response on the keyboard. We used an intermittent tACS protocol (cf. [69]), i.e., tACS was turned on and off in each trial. In 2 different tACS conditions, we tested how the timing of the target word relative to tACS modulates accuracy of reporting the target. In both conditions, the target word was presented so that its p-centre occurred at $3 + y$, $4 + y$, or $5 + y$ seconds after tACS onset, chosen pseudorandomly in each trial (red lines in Fig 4A). $y$ corresponds to 1 out of 6 tested phase delays between tACS and the perceptual centre of the target word, covering 1 cycle of the 3-Hz tACS (corresponding to temporal delays between 66.67 ms and 344.45 ms, in steps of 55.56 ms). In the pretarget tACS condition, tACS was turned off $y$ seconds before the presentation of the target word. In the ongoing tACS condition, tACS remained on during the presentation of the target word and was turned off $1 - y$ seconds after target presentation. In each trial, the background noise was faded in with a random delay relative to tACS onset (between 0 and 0.277 second). This ensured that the interval between noise onset and target was unrelated to the phase lag between tACS and target, avoiding potential alternative explanations for the hypothesized phasic modulation of word report by tACS. The background noise was faded out $1.5 - y$ seconds after target presentation.

Participants completed 10 blocks of 36 trials each, leading to a total of 10 trials for each combination of conditions (tACS condition × duration × phase delay). Prior to the main experiment, they completed a short test in which the signal–noise ratio (SNR) between target word and background noise was adjusted and word report accuracy was assessed. During this test, no tACS was applied. Acoustic stimulation was identical to that in the main experiment, apart from the SNR, which was varied between −8 dB and 8 dB (in steps of 4 dB; 15 trials per SNR). From this pretest, a single SNR condition at the steepest point on the psychometric curve (word report accuracy as a function of SNR) was selected and used throughout the main experiment (methods used for quantification of word report accuracy are described below in Quantification and Statistical analysis). This SNR was, on average −1.05 dB (SD: 1.75 dB).

For those participants who completed both experiments, Experiment 1 was always completed prior to Experiment 2, with, on average, 23 days between experiments (std: 30.88 days). However, all but 2 participants completed both experiments within 1 week of each other.

## MEG/EEG data acquisition and preprocessing (Experiment 1)

MEG was recorded in a magnetically and acoustically shielded room, using a VectorView system (Elekta Neuromag Oy, Helsinki, Finland) with 1 magnetometer and 2 orthogonal planar gradiometers at each of 102 positions within a hemispheric array. EEG was recorded simultaneously using 70 Ag-AgCl sensors according to the extended 10–10 system and referenced to a

sensor placed on the participant's nose. All data were digitally sampled at 1 kHz and band-pass filtered between 0.03 and 333 Hz (MEG) or between 0.1 and 333 Hz (EEG), respectively. Head position and electrooculography activity were monitored continuously using 5 head-position indicator (HPI) coils and 2 bipolar electrodes, respectively. A 3D digitizer (FASTRAK; Polhemus, Colchester, Vermont, USA) was used to record the positions of the EEG sensors, HPI coils, and approximately 70 additional points evenly distributed over the scalp relative to 3 anatomical fiducial points (the nasion and left and right preauricular points).

Data from MEG sensors (magnetometers and gradiometers) were processed using the temporal extension of Signal Source Separation [71] in MaxFilter software (Elekta Neuromag) to suppress noise sources, compensate for motion, and reconstruct any bad sensors.

MEG/EEG data were further processed using the FieldTrip software [72] implemented in MATLAB (The MathWorks, Natick, Massachusetts, USA).

EEG data were high-pass filtered at 1 Hz and re-referenced to the sensor average. Noisy EEG sensors were identified by visual inspection and replaced by the average of neighbouring sensors. For MEG and EEG data separately, artefacts caused by eye movements, blinks, or heartbeat were extracted using independent component analysis (ICA). ICA was applied to data down-sampled to 150 Hz. ICA components representing artefacts were identified visually and removed from the data at the original sampling rate of 1 kHz. The data were then epoched into trials from −3 seconds (longer condition) or −2 seconds (shorter condition) to +2.5 seconds, relative to the omission of the first word in each sequence (cf. Fig 1C).

## Electrical stimulation (Experiment 2)

Current was administered using 2 battery-driven stimulators (DC-Stimulator MR, Neuroconn GmbH, Ilmenau, Germany). Each of the stimulators was driven remotely by the output of 1 channel of a high-quality sound card (Fireface UCX, RME, Germany); another output channel was used to transmit diotic auditory stimuli to the participants' headphones, assuring synchronisation between applied current and presented stimuli.

We used a tACS electrode configuration that has produced a reliable modulation of word report in a previous study [8]. This protocol entails bilateral stimulation over auditory areas using ring electrodes (see inset of Fig 4A). Each pair of ring electrodes consisted of an inner, circular electrode with a diameter of 20 mm and a thickness of 1 mm, and an outer, "doughnut-shaped," electrode with an outer and inner diameter of 100 and 75 mm, respectively, and a thickness of 2 mm. The inner electrodes were centred on T7 and T8 of the 10–10 system, respectively. The parts of the outer electrodes which overlapped with participants' ears were covered using electrically isolating tape. Electrodes were kept in place with adhesive, conductive ten20 paste (Weaver and Company, Aurora, Colorado, USA). Stimulation intensity was set to 1.4 mA (peak-to-peak) unless the participant reported stimulation to be unpleasant, in which case intensity was reduced (consequently, 2 participants were stimulated with 1.2 mA, one with 1.1 mA, and one with 1.0 mA). Current was not ramped up or down; we verified in preliminary tests that for sinusoidal stimulation, this does not lead to increased current-induced sensations.

Sham stimulation was not applied in this experiment. Sensations produced by tACS are typically strongest at the onset of the electrical stimulation. Based on this notion, during sham stimulation, current is usually ramped up and down within several seconds, leading to similar sensations as during "true" tACS, but with no stimulation in the remainder of the trial or block (e.g., [73]). In the current experiment, we tested whether tACS applied for only several seconds leads to a phasic modulation of perception. Given the similarity of this approach to a typical sham stimulation condition, we did not expect that it would act as an appropriate control. Instead, we compared the observed tACS-induced modulation of speech perception with

that obtained in a surrogate distribution, reflecting the null distribution (see Quantification and Statistical analysis).

We verified in pretests that turning on or off the electric stimulation does not produce any sensation that is temporally so precise that participants can distinguish the 2 conditions (note that tACS is applied intermittently in both conditions, only with different timings relative to the target word). However, we did not measure potential sensations quantitatively during the experiment to avoid drawing attention to the transient nature of our tACS protocol. However, even if tACS sensations differed between the 2 conditions at the relevant time points (e.g., during target presentation), they seem unlikely to have affected the hypothesized phasic modulation of word report (for this to happen, participants would also need to distinguish different tACS phases and relate these phases to the time at which the target is presented; see [8] for further discussion). Rather, we might expect a generic effect of tACS such as a difference in overall word report accuracy (averaged across phase). This result was not observed in the current study, and hence we feel confident that the phasic effects of pretarget tACS are due to entrainment of underlying neural mechanisms.

## Statistical analyses

All analyses were implemented using custom MATLAB scripts and the toolbox for circular statistics [74], where appropriate.

**Experiment 1.** We first quantified rhythmic responses in our data using ITC (Fig 1D). At a given frequency and time, ITC measures the consistency of phase across trials [75,76]. ITC ranges between 0 (no phase consistency) and 1 (perfect phase consistency). Although some studies used spectral power to quantify oscillatory activity in rhythmic paradigms (e.g., [2]), ITC can be considered more appropriate in our case as it (1) as a measure based on phase, not power, directly takes into account the temporal structure of the data [20] and (2) is less affected by power differences across trials, which can bias results (e.g., trials with disproportionally high power can dominate the outcome). ITC at frequency $f$ and time point $t$ was calculated as follows:

$$ITC(f, t) = \left| \frac{1}{N} \sum_{n=1}^{N} e^{i(\varphi(f,t,n))} \right|$$

where $\varphi(f,t,n)$ is the phase in trial $n$ at frequency $f$ and time point $t$, and $N$ is the number of trials.

$\varphi$ was estimated using FFT in sliding time windows of 1 second (step size 20 ms; shown in grey in Fig 1C and 1D), leading to a frequency resolution of 1 Hz. Note that, when the outcome of this time-frequency analysis is displayed (Figs 1E, 3C, 5A, 5B, 5E and 6B), "time" always refers to the centre of this time window.

ITC was calculated separately for each of the 204 orthogonal planar gradiometers and then averaged across the 2 gradiometers in each pair, yielding 1 ITC value for each of the 102 sensors positions. Data from magnetometers were only used for source localization (see below).

Our hypothesis states that we expect stronger rhythmic responses (i.e., ITC) at a given frequency when it corresponds to the rate of the (preceding) stimulus sequence (I and III in Fig 1E and 1F) than when it does not (II and IV in Fig 1E and 1F). We developed an index to quantify this rate-specificity of the measured brain responses (RSR). An RSR larger than 0 reflects a rhythmic response which follows the stimulation rate:

$$RSR_t = (ITC(f = 2, r = 2, t) - ITC(f = 2, r = 3, t)) +$$
$$(ITC(f = 3, r = 3, t) - ITC(f = 3, r = 2, t))$$

where *f* and *r* correspond to the frequency for which ITC was determined and sequence rate (both in Hz), respectively. For most analyses, *t* corresponds to a time interval within which ITC was averaged. Two such intervals were defined (white boxes in Fig 1E): one to quantify RSRs during the sequences, but avoiding sequence onset and offset (−1 to −0.5 second relative to the first omitted word), termed "entrained." The other to quantify RSRs that outlast the sequences, and avoiding their offset (0.5 to 2 seconds relative to the first omitted word), termed "sustained."

To test whether rhythmic responses are present in these time windows and in the different conditions, we compared the RSR against 0, using Student *t* test (one-tailed, reflecting the one-directional hypothesis). We used two-tailed repeated-measures *t* tests to compare RSR between intelligible and unintelligible conditions (16-channel versus 1-channel speech, averaged across durations), between shorter and longer sequences (2 seconds versus 3 seconds, averaged across intelligibility conditions), and to test for their interaction (by comparing their difference). In experimental designs with 2 conditions per factor, this approach is equivalent to an ANOVA. For all sensors and conditions (intelligibility, duration) separately, we verified that the RSR is normally distributed ($p > 0.05$ in Kolmogorov–Smirnov test), a prerequisite for subjecting it to parametric statistical tests. Note that such a behaviour is expected, given the central limit theorem (combining multiple measures leads to a variable that tends to be normally distributed). S5A Fig shows the distribution of RSR, averaged across sensors and conditions. Finally, we constructed a surrogate distribution to verify that an RSR of 0 indeed corresponds to our null hypothesis. This was done by adding a random value to the phase in each trial before recalculating ITC and RSR as described above, and repeating the procedure 100 times to obtain a simulated distribution of RSR values in the absence of a rhythmic response. This distribution of RSR values was indeed centred on 0, and its 95% confidence interval included 0 (S5B Fig). Once again, this justifies our use of parametric statistical tests to confirm whether the observed RSR is greater than zero.

Statistical tests were applied separately for each of the 102 MEG sensor positions (i.e., gradiometer pairs; Fig 2). Significant RSR (differences) were determined by means of cluster-based permutation tests (5,000 permutations) [77]. Sensors with a *p*-value < = 0.05 were selected as cluster candidates. Clusters were considered significant if the probability of obtaining their cluster statistic (sum of t-values) in the permuted dataset was < = 5%.

Electro- or neurophysiological data analysed in the spectral domain (e.g., to calculate ITC) often include aperiodic, nonoscillatory components with a "1/f" shape [23,24]. These 1/f components can bias the outcome of spectral analyses [23,24]. Although this primarily affects estimates of oscillatory power (e.g., higher power for lower frequencies), higher power leads to more reliable estimates of phase and therefore potentially also to higher ITC (even though this measure is analytically independent of power, see above). 1/f components are also influenced by stimulus input [78]. Consequently, it is possible that these aperiodic components differ between stimulus rates and therefore affect our RSR. To rule out such an effect, we repeated our RSR analysis, using ITC values corrected for 1/f components. For this purpose, a 1/f curve [24] was fitted to the ITC as a function of neural frequency, averaged within the time window of interest (dashed lines in Fig 3B, left). This was done separately for each participant, sensor, stimulus rate, and experimental condition (intelligibility and duration), as these factors might influence the shape of the aperiodic component. Each of these fits was then subtracted from the corresponding data; the resulting residuals (Fig 3B, right) reflect 1/f-corrected ITC values and were used to calculate RSR as described above. This procedure revealed prominent peaks at neural frequencies corresponding to the 2 stimulus rates (Fig 3B, right), suggesting successful correction for aperiodic, nonoscillatory components. Given the absence of a pronounced 1/f component in the entrained time window (Fig 3A), we here only show results for the sustained time window (Fig 3B, S1 Fig).

Participants' sensitivity to detect an irregularity in the stimulus rhythm was quantified using d-prime (d′), computed as the standardized difference between hit probability and false alarm probability:

$$d' = z(p_{hit}) - z(p_{false\ alarm})$$

where, in a given condition, $p_{hit}$ and $p_{false\ alarm}$ are the probability of correctly identifying an irregular sequence and falsely identifying a regular sequence as irregular, respectively.

To test whether performance in this task is correlated with rate-specific brain responses during or after the rhythmic sounds, we selected MEG sensors which responded strongly in the 2 time windows defined. In the entrained time window, all sensors were included in a significant cluster revealed by the analyses described above (Fig 2C); we therefore selected the 20 sensors with the largest RSR. In the sustained time window, we selected all sensors which were part of a significant cluster (Fig 2F). The RSR from those sensors (averaged within the respective time window) was correlated with performance (d-prime), using Pearson correlation. Even in conditions with relatively weak brain responses, these can still be related to task performance. For the correlation analysis, we therefore averaged both RSR and d-prime across conditions (intelligibility, duration, and rate, the latter for d-prime only).

MEG analyses in source space are not necessarily superior to those in sensor space, in particular when the signal of interest is expected to be relatively weak [79], such as in the current study (rhythmic brain responses in the absence of sensory stimulation). While sensor space analyses are assumption free, reconstruction methods required for transformation to source space all make certain assumptions which can lead to increased uncertainty if they are invalid [80]. Given that we do not require inferences about the exact spatial location or extent of the hypothesized sustained oscillations, we focus here on analyses in sensor space. Nevertheless, we do also report results in source space for completeness, while emphasising that they should be, for these reasons, be interpreted with caution.

RSR measured with MEG were source localized using the following procedure. First, for each participant, MEG data was coregistered with their individual T1-weighted structural MRI, via realignment of the fiducial points. A structural MRI scan was not available for 1 participant, who was excluded from source analysis. Lead fields were constructed, based on individual MRI scans, using a single-shell head model. Brain volumes were spatially normalized to a template MNI brain and divided into grid points of 1 cm resolution. Source reconstruction was then performed, using a linear constrained minimum variance beamformer algorithm (LCMV [81]). Spatial filters were estimated, one for each of the 2 time windows of interest (entrained and sustained), and for each of the 2 neural frequencies that contribute to the RSR (2 Hz and 3 Hz). For each spatial filter, data from the 2 stimulus rates (2 Hz and 3 Hz) were combined, and single trials were band-pass filtered (second order Butterworth) at the frequency for which the filter was constructed (2 Hz filter: 1 to 3 Hz; 3 Hz filter: 2 to 4 Hz). Data from gradiometers and magnetometers were combined. To take into account differences in signal strength between these sensor types, data from magnetometers were multiplied by a factor of 20 before the covariance matrix (necessary for LCMV beamforming) was extracted. Using other factors than 20 did not change results reported here. The spatial filters were then applied to fourier-transformed single-trial data at the frequency for which the filters were constructed (2 Hz and 3 Hz). The spatially filtered, fourier-transformed single-trials were then combined to form ITC, using the formula provided above. For each of the 2 stimulus rates (2 Hz and 3 Hz), this step yielded 1 ITC value per neural frequency of interest (2 Hz and 3 Hz), and for each of 2,982 voxels inside the brain. These ITC values were then combined to RSR values, as described above.

**Experiment 2.** Participants' report of the target word was evaluated using Levenshtein distance [82], which is the minimum number of edits (deletions, insertions, etc.) necessary to change a phonological representation of the participants responses into the phonology of the target word, divided by the number of phonemes in the word. Accuracy in the task was defined as 1 –Levenshtein distance; this measure varies between 0 and 1, where 1 reflects a perfectly reproduced target word (see [25] for details).

For each participant, tACS condition, and duration separately, we tested how report accuracy varies with phase lag (corresponding to the delay between target word and tACS offset in the pretarget tACS condition and to the actual tACS phase in the ongoing tACS condition; see Fig 4A). This was done by fitting a cosine function to task accuracy as a function of phase lag (Fig 4D), an approach which has recently been revealed as highly sensitive at detecting a phasic modulation of perception [21]. The amplitude of the cosine ($a$ in Fig 4D) reflects how strongly performance varies as a function of phase lag. Note that $a$ is always larger than 0. To test statistical significance, we therefore constructed a surrogate distribution, which consists of amplitude values that would be observed in the absence of the hypothesized phase effect. For this purpose, phase lags were randomly assigned to trials and the analysis repeated to these shuffled datasets. This procedure was repeated 1,000 times, yielding 1,000 amplitude values for each experimental condition. The surrogate distribution was then compared with the single outcome obtained from the original, nonpermuted data, resulting statistical (z-) values, according to:

$$z = (d - \mu)/\sigma$$

where d is the observed data, and $\mu$ and $\sigma$ are mean and standard deviation of the surrogate distribution, respectively [21,22].

The phasic modulation of task accuracy, induced by tACS in a given condition, was considered reliable if the z-value exceeded a critical value (e.g., $z = 1.645$, corresponding to a significant threshold of $\alpha = 0.05$, one-tailed). We first tested for a phasic modulation of word report accuracy, irrespective of tACS duration (Fig 4F). For this purpose, data were pooled over tACS duration before the cosine amplitudes were extracted. We then repeated the cosine fit procedure, separately for each duration (Fig 4G). We analysed the data separately for each tACS condition, as well as for their average. For the latter, cosine amplitude values were averaged since this does not require a consistent preferred phase for both conditions. For all statistical tests, values obtained from the surrogate distribution were treated in the same way as described for the original data.

To evaluate differences in phasic modulation of task accuracy between tACS conditions and durations, additional surrogate distributions were constructed by randomly assigning the variable of interest (i.e., tACS condition or tACS duration) to single trials and recomputing cosine amplitudes. To test for differences between tACS conditions, the difference in cosine amplitude between the 2 conditions was compared with the same difference in the surrogate distribution, using z-values as described above (two-tailed). Likewise, to test for differences between tACS durations, for each tACS condition separately and for their average, the difference in cosine amplitude between the longest (5-second) and shortest (3-second) durations was compared with the same difference in the surrogate distribution (one-tailed). To test for an interaction between tACS condition and duration, we first determined the difference in cosine amplitude between 5-second and 3-second tACS for each tACS condition, and then compared the difference between the 2 conditions with the same difference in the surrogate distribution (two-tailed).

**Experiment 1 versus 2.** Given the expected relationship between tACS and EEG [29–32], we tested whether the phase lag between tACS and target word, leading to particularly accurate

or inaccurate responses in Experiment 2, can be predicted from the phase of EEG responses to rhythmic speech sequences in Experiment 1.

For this purpose, at each time point throughout the trial, EEG phase ($\varphi_{EEG}$, green in Fig 5B-II) was extracted at 3 Hz (corresponding to the frequency at which tACS was applied in Experiment 2). Note that $\varphi_{EEG}$ corresponds to $\varphi(f,t)$ defined above, where f = 3 Hz, and phase was averaged across trials at time point $t$. As described above, $\varphi$ was estimated using FFT and sliding analysis windows of 1 second. $\varphi_{EEG}$ can therefore be understood as the phase of a 3-Hz cosine fitted to data within this 1-second window (shaded grey in Fig 5B-I). The value of $\varphi_{EEG}$ corresponds to the distance between each of the 3 cosine peaks and the end of the corresponding cycle (defined as π; arrow in Fig 5B-I).

To obtain a more reliable estimate of phase, we combined phase estimates within each of the 2 time windows of interest (entrained and sustained). As averaging $\varphi_{EEG}$ across time would lead to phase cancellation effects, we first determined, for each time point, the phase relation (i.e., circular difference) between EEG and the presented sequences. For the latter, $\varphi_{Sound}$ (orange in Fig 5B-II) was defined so that the perceptual centre of each word corresponds to π (compare example sounds on top of Fig 5B-I with $\varphi_{Sound}$ in Fig 5B-II). Assuming a rhythmic EEG response that follows the presented sounds, the phase lag between $\varphi_{EEG}$ and $\varphi_{Sound}$ should be approximately constant across time. The circular difference between the two, labeled $\varphi_{EEGvsSound}$ (Fig 5B-III) was therefore averaged within each of the 2 time windows. For the longer (3-second) sequences in Experiment 1, the entrained time window was extended to −2 to −0.5 second relative to the first omitted word (−1 to −0.5 second for shorter sequences).

For each the 2 tACS conditions, the phase of the cosine fitted to individual data, averaged across durations, was extracted ($\varphi_{tACS}$ in Fig 4D). $\varphi_{tACS}$ reflects the position of the cosine peak (i.e., the "preferred" tACS phase, leading to highest accuracy), relative to the maximal phase lag tested (here: π).

For each participant, EEG electrode, and combination of conditions in the 2 experiments, we then extracted the circular difference between $\varphi_{tACS}$ (Fig 4D and 4E) and $\varphi_{EEGvsSound}$ (Fig 5B-III and 5C-I). The distribution of this difference (Fig 5C-II and 5C-III) reveals whether there is a consistent phase lag between $\varphi_{tACS}$ and $\varphi_{EEGvsSound}$ across participants. In this case, we would expect a nonuniform distribution, which was assessed with Rayleigh test for nonuniformity (Fig 5D). Despite potential differences in the magnitude of rhythmic brain responses, the different sequence durations tested in Experiment 1 should not differ in their phase relation to the sound. The $\varphi_{EEGvsSound}$ obtained in these conditions were therefore averaged. Finally, we selected 29 EEG sensors whose phase during intelligible speech was predictive (FDR-corrected $p < = 0.05$ in Rayleigh test) for $\varphi_{tACS}$ in the pretarget tACS condition (cf. Fig 5D). The z-values, obtained from Rayleigh test, were averaged and displayed as a function of time (i.e., not averaged within the 2 windows as described above).

Although methodologically more distant to tACS than EEG (only the latter two are affected by distortions by skull and tissue), we repeated the procedure for the simultaneously acquired MEG data (S3 Fig). Here, to avoid phase cancellation effects, z-values were calculated separately for each of the 204 gradiometers and then averaged across the 2 gradiometers in each pair, yielding 1 z-value for each of the 102 sensors positions (note that z-values from Rayleigh test are always larger or equal to 0).

We also used the obtained results to realign behavioural outcomes in Experiment 2 relative to the predicted optimal tACS phase (leading to highest accuracy) in individual participants. The primary purpose of this realignment is to illustrate implications of results obtained in the analysis described in the preceding paragraph (Fig 5D). We also used a leave-one-participant-out procedure to avoid the inherent circularity in defining preferred phases or phase lags with the same data as used in the eventual analysis. This procedure is depicted in Fig 6.

Step 1 (Fig 6A): For each participant $i$, data from all remaining participants were used to estimate the average difference between $\varphi_{tACS}$ (from the pretarget tACS condition) and $\varphi_{EEGvsSound}$. $\varphi_{EEGvsSound}$ was determined in the entrained time window, at electrode F3 (showing the highest predictive value for $\varphi_{tACS}$ in the pretarget condition). Step 2 (Fig 6B): $\varphi_{EEGvsSound}$ was determined for participant $i$. Step 3 (Fig 6C): The $\varphi_{EEGvsSound}$, obtained for participant $i$ in step 2, was shifted by the average difference between $\varphi_{tACS}$ and $\varphi_{EEGvsSound}$, obtained in step 1. This yielded the predicted $\varphi_{tACS}$ for participant $i$. Step 4 (Fig 6D): The predicted $\varphi_{tACS}$ was used to estimate the tACS phase lag with highest perceptual accuracy for participant $i$. This phase lag was calculated as $\pi$-$\varphi_{tACS}$, based on the fact that $\varphi_{tACS}$ reflects the distance between the peak of a fitted cosine and the maximal tACS phase lag (Fig 4B). The behavioural data from participant $i$ was then shifted by the predicted optimal phase lag, so that highest accuracy was located at a centre phase bin. As behavioural data were only available for 6 different phase lags, it was (linearly) interpolated between these data points (167 interpolated values between each phase lag) to enable a more accurate realignment of the data (note that the predicted $\varphi_{tACS}$ depends on (1) the phase of the cosine fitted to individual data and (2) $\varphi_{EEGvsSound}$, neither of which are restricted to the 6 phase values tested). Step 5 (Fig 6E): Steps 1 to 4 were repeated, separately for each of the 18 participants. Step 6 (Fig 6F): The realigned data were averaged across participants, with the hypothesis of highest accuracy at the predicted optimal phase lag for word report accuracy. This hypothesis was tested by comparing accuracy at this phase lag (0 in Fig 6F) with accuracy at the one 180˚ (or $\pi$) away, using a one-tailed (given the clear one-directional hypothesis) paired $t$ test.

In a final analysis, we used this realignment procedure to test whether a modulation of perception during or after tACS reflects enhancement or disruption of perception (or both). As our experimental protocol prevented the inclusion of the usual sham stimulation condition (see Electrical stimulation), we based this analysis on the finding that $\varphi_{tACS}$ was not reliably predicted by $\varphi_{EEGvsSound}$ in the ongoing tACS condition. We repeated the procedure described in the preceding paragraph; however, we used it to realign behavioral outcome from the ongoing tACS condition to the phase lag predicted to be optimal for word report accuracy. Consequently, the only difference to the procedure described above is the use of $\varphi_{tACS}$ obtained in the ongoing (not pretarget) tACS condition.

We compared accuracy at the predicted optimal tACS phase lag between the 2 tACS conditions. Given that $\varphi_{EEGvsSound}$ is not predictive for $\varphi_{tACS}$ in the ongoing tACS condition, any tACS-dependent changes in perception should be abolished by the realignment procedure, and the outcome reflects the null hypothesis. Consequently, higher accuracy at the predicted optimal phase lag in the pretarget tACS condition indicates an enhancement of speech perception, produced by tACS. This was tested by means of a one-tailed (given the clear one-directional hypothesis) paired $t$ test. Finally, we repeated the alignment procedure for both conditions, but this time aligned the behavioural data at the predicted worst phase lag for speech perception (i.e., 180˚ or $\pi$ away from the predicted optimal phase). Again, we compared accuracy at this predicted worst phase lag between the 2 tACS conditions, using a one-tailed repeated-measures $t$ test. Lower accuracy at the predicted worst phase lag in the pretarget tACS condition indicates a disruption of speech perception, produced by tACS.

## Supporting information

**S1 Fig. RSRs in sustained time window after correction for 1/f component.** Same as in Fig 2D–2F, but using 1/f-corrected ITC (shown in Fig 3B) to calculate RSR. Same conventions as for Fig 2. Please refer to S1 Data for the numerical values underlying this figure. ITC, intertrial coherence; RSR, rate-specific response.
(EPS)

**S2 Fig. Correlation between RSR in the entrained (left) and sustained (right) time windows (for the selected sensors shown in Fig 2C and 2F), respectively, and performance in the irregularity detection task (cf. Fig 1B).** Both RSR and performance were averaged across intelligibility and duration conditions; in addition, performance was averaged across rates. Shaded areas correspond to the confidence intervals of the regression lines. Please refer to S1 Data for the numerical values underlying this figure. RSR, rate-specific response.
(EPS)

**S3 Fig. Using MEG responses to predict optimal tACS phase.** Same as Fig 5D, but using MEG instead of EEG data from Experiment 1. Please refer to S1 Data for the numerical values underlying this figure. EEG, electroencephalography; MEG, magnetoencephalography; tACS, transcranial alternating current stimulation.
(EPS)

**S4 Fig. Data from all individual participants, realigned to predicted optimal tACS phase.** Same as Fig 6D and 6E, but for all 18 participants who were included in the analysis. Note that the average across participants is shown in Fig 6F. tACS, transcranial alternating current stimulation.
(EPS)

**S5 Fig. Control analyses validating RSR as an appropriate measure to reveal rate-specific rhythmic brain responses.** (**A**) Distribution of RSR over participants. Note the approximate normal distribution as required for parametric tests (e.g., $t$ test against 0). Please refer to S1 Data for the numerical values underlying this figure panel. (**B**) Distribution of RSR, averaged across participants, in a surrogate dataset (see Materials and methods). RSR is centred on 0 (dashed lines), validating our null hypothesis of RSR = 0. For all results shown here, RSR values have been averaged across sensors and conditions (corresponding to the average RSR shown in Fig 2A and 2D), including those for which the RSR is not reliably different from 0. Statistically significant RSRs after intelligible speech are shown in Fig 2F. Note that x-axes are not identical across panels. RSR, rate-specific response.
(EPS)

**S1 Data. Excel spreadsheet containing, in separate sheets, the underlying numerical data for figure panels 1B, 2A–2F, 3A–3C, 4E–4G, 5A, 5C–5E, 6F, 6G, S1A–S1D, S2, S3, and S5A.**
(XLSX)

## Acknowledgments

The authors thank Isobella Allard for support during pilot testing, Loes Beckers and Clare Cook for help with data acquisition, and Anne Kösem, Nina Suess, and Nathan Weisz for advice on MEG source localization.

## Author Contributions

**Conceptualization:** Matthew H. Davis, Benedikt Zoefel.

**Formal analysis:** Sander van Bree, Benedikt Zoefel.

**Funding acquisition:** Matthew H. Davis, Benedikt Zoefel.

**Investigation:** Sander van Bree.

**Methodology:** Sander van Bree, Ediz Sohoglu, Matthew H. Davis, Benedikt Zoefel.

**Project administration:** Benedikt Zoefel.

**Software:** Benedikt Zoefel.

**Supervision:** Matthew H. Davis, Benedikt Zoefel.

**Visualization:** Matthew H. Davis, Benedikt Zoefel.

**Writing – original draft:** Benedikt Zoefel.

**Writing – review & editing:** Sander van Bree, Ediz Sohoglu, Matthew H. Davis, Benedikt Zoefel.

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
