## [Editor Report · Decision Letter 0]

6 Aug 2020

Dear Dr Zoefel, 

Thank you for submitting your manuscript entitled "Sustained neural rhythms reveal endogenous oscillations supporting speech perception" for consideration as a Research Article by PLOS Biology.

Your manuscript has now been evaluated by the PLOS Biology editorial staff as well as by an academic editor with relevant expertise and I am writing to let you know that we would like to send your submission out for external peer review.

Please re-submit your manuscript within two working days, i.e. by Aug 10 2020 11:59PM.

Kind regards,

Lucas Smith, Ph.D.,

Associate Editor

PLOS Biology

---

## [Decision Letter · Decision Letter 1]

29 Sep 2020

Dear Dr Zoefel,

Thank you very much for submitting your manuscript "Sustained neural rhythms reveal endogenous oscillations supporting speech perception" for consideration as a Research Article at PLOS Biology. Your manuscript has been evaluated by the PLOS Biology editors, an Academic Editor with relevant expertise, and by several independent reviewers.

The reviews of your manuscript are appended below. As you will see from their comments, all of the reviewers felt that this is study addresses an important topic, however, each reviewer has raised a number of specific concerns. These include concerns raised by Reviewer 1 that the sustained response seems very weak and does not correlate with behavior. Reviewers 1 and 2 also find it surprising that the modulatory effect of tACS seems to be stronger in the pre-target tACS condition than in the ongoing tACS condition and would like this discussed. Reviewer 3 has concerns regarding the statistical approaches used, which need to be addressed. Moreover, all three reviewers question why tACS data was only correlated with EEG data, but not with the MEG data.

Having discussed the reviews with the Academic Editor, we will not be able to accept the current version of the manuscript, but we would welcome re-submission of a much-revised version that takes into account the reviewers' comments. We cannot make any decision about publication until we have seen the revised manuscript and your response to the reviewers' comments. Your revised manuscript is also likely to be sent for further evaluation by the reviewers.

We expect to receive your revised manuscript within 3 months. However, please let us know if more time would be needed to thoroughly address the reviewer concerns, as it is important that a strong revision be submitted and we would be open to extending the time provided.

**IMPORTANT - SUBMITTING YOUR REVISION**

*Re-submission Checklist*

*Published Peer Review*

*PLOS Data Policy*

*Blot and Gel Data Policy*

Sincerely,

Lucas Smith, Ph.D.,

Associate Editor,

lsmith@plos.org,

PLOS Biology

REVIEWS:

Reviewer's Responses to Questions

PLOS authors have the option to publish the peer review history of their article (what does this mean?). If published, this will include your full peer review and any attached files.

Reviewer #1: No

Reviewer #2: No

Reviewer #3: No

Reviewer #1: The study by van Bree et al. investigated whether endogenous neural oscillations played a role in speech perception and whether they were related to the neural responses entrained to speech. The authors found that (1) the neural responses to rhythmic speech stimuli could last for ~1-2 seconds after the stimulus offset and (2) the phase of neural response to speech could predict how the tACS phase modulated speech perception. Based on these two findings, the authors concluded that the neural responses to speech were related to spontaneous neural oscillations.

In general, the study addressed an important question regarding to the neural mechanisms underlying speech perception. The methods were sophisticated and well described. The tACS results were quite interesting. Nevertheless, I have some concerns about whether the findings can indeed support the conclusion.

First, the authors show some evidence that the entrained neural response could last for 1-2 seconds after the stimulus. Nevertheless, as the authors pointed out, the entrained response and the sustained response are generated from different neural sources, suggesting that the entrained response is not "endogenous neural oscillations aligned (or "entrained") to the stimulus rhythm".

Second, the sustained response does not correlate with behaviour while the entrained response does, providing further evidence that the sustained response does not directly contribute to speech perception. 

Third, I have some doubts about whether the MEG sustained response indeed exist. First of all, it is a very weak response, not observable in the raw ITC spectrogram. Even in Fig. 2H, the 2 and 3 Hz ITC peaks are not the most prominent peaks in the relatively narrow frequency range being shown. Even if the sustained response is statistically significant, it is way smaller than the entrained response. Based on Fig. 2D and Fig. 2Hz, the ITC for entrained response is >0.2 while the ITC for sustained response is <0.12, with the chance-level ITC being above 0.1. Furthermore, the sustained response is not observed for EEG.

In sum, the sustained response seems to be a very weak response that does not correlate with behaviour.

I also have some concerns about the tACS results, which I find interesting. I have some trouble understanding why the tACS has to stop before the target to influence the target detection performance. Based on the MEG results, the sustained response is way weaker than the entrained response, which seems to predict a much stronger effect for "ongoing" tACS.

Furthermore, the tACS phase is correlated with the phase of the evoked EEG response, instead of the phase of the sustained EEG response. Nevertheless, the tACS effect is observed after the tACS stimulation. In other words, the tACS phase is likely to reflect the phase of the sustained response caused by tACS. Therefore, I have trouble understanding why the phase of sustained tACS response is correlated with the phase of entrained EEG response instead of the phase of the sustained EEG response.

Additionally, the correlation between tACS phase and EEG phase is interesting, but I wonder why EEG instead of MEG is used in this analysis. MEG and EEG were simultaneously recorded. However, the MEG data were used to show the existence of a sustained response while the EEG data were used to correlate with the tACS phase. Please explain why the MEG and EEG data were used for different purposes.

Reviewer #2: The manuscript investigates whether neural entrainment to speech and to tACS sustains after the cessation of the stimulation. This observation would be key evidence that observed neural entrainment involves the recruitment of endogenous neural oscillations. For this, two experiments were performed with the same participants : one MEG/EEG experiment and one tACS experiment. The results show that the neural entrainment to intelligible speech outlasts the stimulus. The findings also show that tACS affects perception after the end of electrical stimulation, and that there is a correspondence between the EEG phase during entrainment and the tACS phase that leads to most accurate speech perception.

The study provides new important insights into the mechanisms underlying neural entrainment. It reports an extensive amount of data that are thoroughly analyzed. I find the main results convincing and particularly exciting. I still have a few questions mostly on the tACS results. 

- What I find most surprising in the tACS results is that the modulatory effect seems to be stronger in the "pre-target" tACS condition than in the "ongoing" TACS condition. I think this point needs to be more extensively discussed in the manuscript, especially with regards to past work on tACS-phase effects in auditory/ speech processing. 

I am confused in particular by the discussion section, e.g. p. 659-660:"Our finding of enhanced speech perception therefore supports the hypothesis that tACS can enhance neural entrainment if it is applied in the absence of a "competing" entraining stimulus." I do not follow this argument. Ongoing tACS could potentially interfere with neural oscillations if presented at the wrong phase and/or frequency, but should in principle enhance oscillatory activity if presented at the correct phase ? And based on the past literature (Riecke et al., 2018; Zoefel et al.,2018; 2020), I would still expect a phasic modulation of perception and/or neural response during the "ongoing" tACS condition. Do the authors have an explanation for this apparent discrepancy ? 

I think also that the conclusion l.337 that "rhythmic changes in speech perception outlast the period of tACS" seems a bit too strong with regards to the actual behavioral results (i.e. because of the absence of significant results in the "ongoing" tACS condition).

- Because there is no ramping on and off of the stimulation, could the participants feel when the current stopped in the trial ? Could they guess they were in the "pre-target" condition, or in the "ongoing" condition ?

- Have the authors looked at the correlation between MEG phase and tACS data ? I understand that EEG might reflect activity of networks most likely affected by the tACS current, but it could be interesting to correlate tACS and MEG data knowing that the sustained MEG response is more robust.

- Considering that sustained entrainment is not observed for unintelligible speech, would you consider that this phenomenon is restricted to speech processing ? What about non-verbal rhythms, in auditory and other sensory modalities ?

Reviewer #3: I am much in favour of this manuscript. It contains many important data points--obviously collected with great care and analysed by and large with impressive ingenuity--that there is a biophysical reality to neural entrainment and its behavioural corollaries, and that tACS, in a limited way, can perturb this behavioural (i.e., speech comprehension) outcome.

Not much of this study is per se truly new or unexpected, and the senior authors have published extensively on this--yet, given how contentious the claim of neural entrainment as a mechanism still is, a well-done study like this clearly deserves notice. The results demonstrated in Figure 5 to me are the single most important and most new element, but they also thus should raise most robustness checks (see below).

It should also be noted that the Achilles heel of this study is the low N.

(In fact, I would not bet much money on the main result in Figure 5 (and in fact all correlations with behaviour here) replicating. This could most easily be settled, if it were simply -- replicated. Given that we are inmidst a pandemic, this is a problem I concur)

I do think, however, the authors could do more to comfort their readers that the result in Figure 5 is solid: How much does it hinge on the (contentious; cf. Boateng et al.) individual peak-aligning? How much does it depend on a t-test with df=17 being used? How about comparing the entire two "patterns" in opanels F,G instead? and so forth.

In sum, I very much welcome this study and would be happy for it to enter the canon. However, I urge the authors to bolster their case by an extensive series of clean-up in the domains of researcher degrees of freedom (see below) and statistical reporting/analysis. This will help make a much more solid case.would need to be convinced more of key take-homes. I outline a few more points in more detail below.

***

My main problem for the overall conclusion is in fact one of robustness. ~N=20 is not much for any between-participants/generalising inference when it comes to predicting behaviour.

The researcher degrees of freedom in this paper have been immense obviously. The reader would thus need more demonstrations, or at the very least better rationale, to decide how robust key results are against these choices. A key point here is the switch from demonstrating a brain effect using MEG data in Exp 1, but using EEG data for the comparison with behavioural/tACS effects from Exp 2. Why was this done at all? Why should it not be possible to demonstrate the Fig. 5 effect in source space of MEG?

A second point, pervasive throughout the ms., is the choice of sensors (see below) and time windows in statistical analysis.

-- Distributional properties of the new RSR index:

I am all for developing useful indices but demonstrating their theoretical and empirical properties in a supplemental figure is necessary.

Submitting the calculated RSR to an ANOVA sounds dangerous, to say the least. Both ITC and RSR should probably not be submitted to parametric analyses (mean differences, correlations, etc) at all. With the senior author's expertise in permutation statistics, I strongly recommend to make a consistent/overriding switch to such methods more suited to the data

-- Line 216 f: Selecting the sensors with the largest response runs an unnecessary danger of double-dipping/circularity. Your paper and your point in case will be much stronger if you present a principled way of analysing these data. In fact, I strongly suggest to present these analyses entirely in source space. Source space if the obvious forte of MEG. The prime advantage here would not only lie in a higher (if modest) degree of neurofunctional organisation specificity). More importantly, the move to source space in MEG acts as a spatial filter that can should improve SNR and benefit inference at al levels.

-- Line 228: My point about which sensors/signals to pick, the entrained vs sustained response correlations w/ behaviour deserve more attention. I am not sure whether the authors here want to run with a dissociation interpretation? Then the correlations would need to be tested against each other (unlikely to provide evidence for a difference, in fact; z test for paired correlations would only be significant if we assume a very high correlation between RSR from the two time windows). If the goal here is to argue for an absent interaction, though, predicting behaviour from both time windows in a regression model might be an option, showing that both contribute. In all cases, I was unclear what the authors want to convey with this (low-N, between-subjects, and thus notoriously unstable) correlation.

-- line 314f: Again, I was unclear about the rationale of the authors: Is there or is there not a tACS x condition interaction, i.e. what can we conclude about the specificity of the tACS condition/window effect on word accuracy? The three separate z tests also suffer from lower sensitivity due to less data being included, of course.

Minor:

-- reporting only p-values for cluster tests, correlation tests, etc is not state of the art.

-- font size in all but a few figure panels is prohibitively small

-- line 424: Just another illustration of my major point of contention, where the EEG channel choice seems overly tailored to the data. More efforts of choosing sensors/ROIs/time windows independent of the data to be tested on it need to be entertained by the authors.

-- Subheading "Sustained oscillations produced by tACS enhance, but do not disrupt speech perception" does need statistical qualification (i.e. interaction pattern). The authors run danger once more of arguing with not necessarily meaningful differences of differences.

---

## [Decision Letter · Decision Letter 2]

20 Jan 2021

Dear Dr Zoefel,

Thank you for submitting your revised Research Article entitled "Sustained neural rhythms reveal endogenous oscillations supporting speech perception" for publication in PLOS Biology. I have now obtained advice from the original reviewers and have discussed their comments with the Academic Editor. 

Based on the reviews, we will probably accept this manuscript for publication, assuming that you will modify the manuscript to address the remaining points raised by reviewer 1. We have discussed these lingering concerns with the Academic Editor and think they can be simply addressed with textual changes to your manuscript.

***IMPORTANT: Please also address the ethics & data requests included at the bottom of this email before resubmitting***

We expect to receive your revised manuscript within two weeks. Your revisions should address the specific points made by each reviewer. 

-  a cover letter that should detail your responses to any editorial requests, if applicable

*Published Peer Review History*

*Early Version*

Sincerely,

Lucas Smith, Ph.D.,

Associate Editor,

lsmith@plos.org,

PLOS Biology

ETHICS STATEMENT:

--Please indicate whether your protocol, approved by the Cambridge psychology research ethics committee adhered to the Declaration of Helsinki or other national or international ethical guidelines.

-- Please include the ID number of the protocol approved by the Cambridge Psychology Research Ethics Committee.

-- Please note whether the informed consent obtained from participants in this study was oral or written. If consent was oral, please explain why

DATA POLICY:

Note that we do not require all raw data. Rather, we ask that all individual quantitative observations that underlie the data summarized in the figures and results of your paper be made available.

--Please provide (or indicate where in the manuscript this can be found), as a supporting file, a spreadsheet that contains the individual numerical values that were used to generate the summary statistics show in Figure 1B-F; 2A-F; 3A-C; 4D,E-G; 5A-D; 6A-G; Figure S1A-D; S2A-B; S3; S4; S5A-B. For an example spreadsheet see here: http://www.plosbiology.org/article/info%3Adoi%2F10.1371%2Fjournal.pbio.1001908#s5

Please also indicate, within each figure legend, where this underlying data may be found and ensure your supplemental data file/s has a legend.

-- Please include in your deposition in OSF a README file that would allow the reader to link your data files to each of the figures displaying quantitative data, by explaining how the data was analyzed to generate the final plots and graphs.

--Please ensure that your Data Statement in the submission system accurately describes where your data can be found.

Reviewer remarks:

Reviewer's Responses to Questions

PLOS authors have the option to publish the peer review history of their article (what does this mean?). If published, this will include your full peer review and any attached files.

Reviewer #1: No

Reviewer #2: No

Reviewer #3: No

Reviewer #1: The authors have done a good job revising the manuscript. However, I still have a few concerns.

1. "Like other spectral measures, ITC is affected by aperiodic ("1/f") activity, leading to larger ITC for lower frequencies without necessarily involving endogenous oscillatory activity."

ITC certainly shouldn't be affected by the 1/f spectrum. The 1/f shape is for the power spectrum while the ITC analyzes the phase coherence over trials. In fact, the authors should explain why the ITC reported here has a 1/f trend.

2. "The current applied to the scalp during tACS is distorted by skull and tissue before it reaches the brain. The electrical signal produced by the brain is similarly distorted before being captured using EEG electrodes attached to the scalp. Importantly, MEG is not affected by such distortions. Consequently, EEG is methodologically closer to tACS than MEG."

I can't agree with this statement. By this logic, the authors should have stimulated the bilateral auditory cortices by applying tACS at the position of Cz, instead of applying tACS bilaterally. If a signal is conducted from position A to position B, there is no garantee that stimulation on position B will activate position A, and vice versa.

Finally, I still have some concerns about the hypothesis that any neural activity that can last for a couple of seconds reflects endogenous activity. After a hit, a bell can ring for several seconds but I'm not sure if this kind of ringing should be called endogenous.

Reviewer #2: I thank the authors for their thorough answers. They have addressed all my concerns, and I recommend the manuscript for publication.

Reviewer #3: The authors have provided very extensive and compelling re-analyses and have reworked their ms. to standards entirely satisfactory to me/my original concerns.

I suggest its acceptance in Plos Biology as is.

 -- Jonas Obleser

---

## [Editor Report · Decision Letter 3]

8 Feb 2021

Dear Dr Zoefel,

On behalf of my colleagues and the Academic Editor, Christopher Pack, I am pleased to say that we can in principle offer to publish your Research Article "Sustained neural rhythms reveal endogenous oscillations supporting speech perception" in PLOS Biology, provided you address any remaining formatting and reporting issues. These will be detailed in an email that will follow this letter and that you will usually receive within 2-3 business days, during which time no action is required from you. When addressing these last points, please also update the data availability statement in your manuscript to include a reference to the supplementary data file S1 (containing the underlying data for your figures).

For example, your data availability statement may say something like:

Data is available in supplementary file Data S1 and at (https://osf.io/xw8c4/). Custom-built MATLAB scripts are available at (https://osf.io/xw8c4/).

Please note that we will not be able to formally accept your manuscript and schedule it for publication until you have made the required changes.

PRESS

Thank you again for supporting Open Access publishing. We look forward to publishing your paper in PLOS Biology. 

Sincerely, 

Lucas Smith, Ph.D. 

Senior Editor 

PLOS Biology